# The effectiveness of prosocial policies: Gender differences arising from social norms

**Antonio Cabrales[1] \*, Ryan Kendall[2], Angel Sánchez[3,4]**

**1** Departamento de Economía, Universidad Carlos III de Madrid, Getafe, Madrid, Spain, **2** Center for Bioethics, School of Global Public Health, New York University, New York, NY, United States of America, **3** Grupo Interdisciplinar de Sistemas Complejos (GISC), Departamento de Matemáticas, Universidad Carlos III de Madrid, Leganés, Madrid, Spain, **4** Instituto de Biocomputación y Física de Sistemas Complejos (BIFI), Universidad de Zaragoza, Zaragoza, Spain

\* antonio.cabrales@uc3m.es

## Abstract

We study policies aimed at discouraging behavior that produces negative externalities, and their differential gender impact. Using driving as an application, we carry out an experiment where slowest vehicles are the safest choice, whereas faster driving speeds lead to higher potential payoffs but higher probabilities of accidents. Faster speeds have a personal benefit but create a negative externality. We consider four experimental policy conditions: a baseline situation, a framing condition in which drivers are suggested that driving fast violates a social norm, and two punishment conditions, one exogenous and one endogenous. We find that the most effective policies use different framing and endogenously determined punishment mechanisms (to fast drivers by other drivers). These policies are only effective for female drivers which leads to substantial gender payoff differences. Our data suggest that these results arise from differences in social norms across genders, thus opening the way to designing more effective policies.

## Introduction

In this paper, we study the effectiveness of policies aimed at discouraging antisocial behavior. By this, we mean behavior that benefits the individual, but that harms the collective. As described in more detail below, previous literature suggests that such policies could have differential gender effects. In this context, our main question of interest is to measure the differential effect on genders of policies designed to promote cooperative human behavior. Furthermore, we want to understand if those differential effects, if they exist, are related to a different propensity to comply with social norms.

Since social norms are highly sensitive to the situation where they are applied [1, 2], we decided to frame the experiment for a specific kind of antisocial behavior. Thus, we focus on policies to deter unsafe driving. However, there are also other behaviors that share similar features to this one, and hence we feel that the results will have wider relevance. For example, some transportation technologies are (currently) more individually costly, but more climate and environmentally friendly than others (electric, hybrid, and gas-powered vehicles). Just to

**Data Availability Statement:** The full dataset and the codes used for the manuscript are now available at https://doi.org/10.7910/DVN/L1YR71.

**Funding:** This research was funded by a grant from the British Academy (SRG\171072, A.C., R.K.) and

by Ministerio de Ciencia, Innovación y Universidades/FEDER (Spain/UE) (through) grant PGC2018-098186-B-I00 (BASIC, A.S.). The funders had no role in study design, data collection and analysis, decision to publish, or preparation of the manuscript.

**Competing interests:** The authors have declared that no competing interests exist.

mention one other example, wearing a face-mask reduces personal comfort but it also reduces the risk of both personal and generalized infections.

In any event, unsafe driving is an important problem on its own in view of the detrimental negative impacts it has on society. Excessive speed is the number one road safety problem in most countries [3] and men, especially young men, are disproportionately involved in accidents [4]. A recent field experiment quantifies the relationship between speed and negative outcomes [5], such as higher deaths from both accidents and pollution. Indeed, following a 10 mph increase in speed limits, affected freeways experienced a 3–4 mph increase in travel speed which is associated with 9–15 percent more accidents and 34–60 percent more fatal accidents. Furthermore, faster speeds have negative externalities such as elevated concentrations of carbon monoxide (14–25 percent), nitrogen oxides (9–16 percent), ozone (1–11 percent) and higher fetal death rates around the affected freeways (9 percent). Beyond speed, other major causes of accidents are distractions and intoxication. For instance, according to the US Department of Transportation, 11 654 people died in alcohol-impaired driving traffic accidents, while distracted driving claimed 3 142 lives in the US, only in 2020. Both share similar features with speed. They benefit the "user" of the behavior (at least in terms of revealed preference), but they definitely harm others that can be innocent victims of an accident as a result of that behavior. In this context, one reason to worry about antisocial behavior on the road, and about good policies to reduce it, is that the coming wave of automation will make it more important. The increased safety of autonomous vehicles safety might encourage free-riding among other drivers. Finally, it is worth mentioning that recent work has shown that providing information on accidents is distracting and actually increases them due to cognitive fatigue [6], which raises the question as to whether priming may be useful as they do not add to the cognitive fatigue while driving, another of the issues we will adress below.

Our research is related to a long tradition of studying behavior in social dilemmas, in particular from an experimental viewpoint. Unsurprisingly, punishment is highly effective in enforcing social norms (see the pioneering work in [7, 8], or [9–13]). A recent field experiment shows that external punishment (along with monitoring) can decrease bribing behavior in education [14]. In addition, moral suasion is also a powerful mechanism to develop social norms around prosocial behavior. Another field experiment testing the policy effectiveness in the domain of energy demand shows that the combination of moral suasion and economic incentives produce substantially different policies [15].

On the other hand, there are interesting, but inconsistent, differences along gender lines relating to prosocial behavior. Female participants are more averse to inequality [16] and less likely to lie or cheat for monetary benefit [17, 18]. In this respect, we note that whether there are gender differences in lying has been largely debated and is a complicated issue. See [19] for a meta-analysis of how gender aligns with different types of lying. Male participants are more likely to violate the social norm when they can do so privately [20]. In addition, evidence has suggested that the neural correlates for social norm compliance are systematically different across genders [21]. In spite of this, when analyzing situations closest to our study, there is plenty of mixed evidence pertaining to the level of prosociality between genders. For example, a review of public goods experiments shows that gender differences are not straightforward and that the context plays a crucial role [22]. A similar nuanced story is shown for punishment to free riders [23] and for charitable donations [24]. Female participants are more prone to donate in dictator games when it is more costly to themselves, whereas male participants donate more when it is cheap [25]). Finally, whether it is true or not, participants expect female participants to be more altruistic than male participants [26].

Given that our work focuses on the specific frame of unsafe driving, we should note that there is an extensive literature in transportation science that identifies the effect of gender on

driving behavior. Oviedo-Trespalacios *et al.* [27] shows that males are distracted more often by roadside signs but females look at them longer. Li *et al.* [28] find that women are less aggressive when encountering a yellow light. Interestingly for our research, Jorgensen and Polak [29] find that although males drive faster in the absence of speeding fines, there is no gender difference when fines are present. More recent research by Elias [30] also finds no difference in gender attitudes to speeding fines. A difference with our framework is that in the latter two cases the authors consider environments where less speeding by one group does not makes it easier for others to speed.

Policy levers that discourage unsafe driving behavior can have immense societal benefits. A central concern for people focused on driving safety is to understand what type of incentive is effective in this setting. Unsurprisingly, there is a close relationship between prosocial driving behavior and exogenous punishment. For example, a 35 percent decrease in roadway troopers was accompanied with a decrease in citations and a significant increase in injuries and fatalities [31]. Fines are particularly effective to deter traffic violations by women [32]. Endogenous social pressures also have an impact on driving behavior. Drivers in Tsingtao, China had less traffic violations when they received text messages with comparisons of other driving behaviors within, and outside of, the social group [33]. Endogenous intra-group pressure is particularly effective in enforcing a social norm. For example, a study in Kenya shows that placing messages inside long-distance minibuses encouraging passengers to speak up against unsafe driving reduced insurance claims by one-half to two-thirds [34]. In addition, previous studies have shown that "males, on average, felt less confident in their ability to influence other drivers and perceived more costs in doing so than females did" [35].

In order to gain further insight on this issue, namely gender differences in the effects of policies to deter unsafe driving, in this paper we introduce a scenario where drivers choose between two manual driving styles ("Fast" or "Slow") and one style which allows their vehicle to drive automatically ("Auto"). We assume that Auto drivers are never in an accident and will therefore earn a constant amount. Both Fast and Slow have positive probabilities of accidents, and higher speed than Auto (higher speed brings a higher payoff). Fast is faster and riskier, and Slow is slower and less risky. In addition, faster average driving speeds also increase the probability that *all* non Auto drivers are involved in an accident (although Fast is more likely to be involved in one than Slow), thus creating a negative externality on the population. In such a context, policies can play a role in discouraging individual drivers from free-riding off of the safety provided by others' safer driving styles. We used this scenario to design a laboratory experiment where we could test different policy interventions building on our knowledge of human cooperation, namely incorporating ideas of framing and punishment. As we will show below, we have found that no policy has an effect on the average population in terms of reducing the most dangerous driving style, Fast. However, this average null effect hides a remarkable asymmetry. All policy conditions have the same (expected) effect on female participants—they choose Fast less often and Auto more often. However, males now choose Fast more often, taking advantage of the safer environment created by the choices of females. This non trivial gender difference requires an explanation and we will discuss the mechanisms in detail.

## Materials and methods

### Game

$N$ participants play a game in which each one chooses a driving style $D_i \in \{F, S, A\}$, ($F$ stands for *Fast*, $S$ stands for *Slow*, and $A$ for *Auto*mated). The payoff of each player depends on the speed of each strategy $S_i$, which we assume is ordered so that $S_F > S_S > S_A > 0$. It also depends

on the probability of having an accident. That probability increases with the speed of the strategy chosen, and with the average speed chosen by all the players. Letting $x_j$ denote the proportion of other players choosing strategy $j$, then the Average Speed of the players other than $i$ in the game is given by

$$AS_{-i} = x_F S_F + x_S S_S + x_A S_A$$

and $p_i$ the probability that a player choosing strategy $i$ is involved in an accident, which is affected by average speed $AS_{-i}$ and an idiosyncratic factor to the strategy $a_i$. That is, the probability increases in the average speed, but it is proportionally higher for more "aggressive" strategies. The kind of accident we are thinking about is one where the driver causing it is involved, so they are partially "punished" by losing their payoff, but there is no extra punishment. This is realistic because we are thinking of $F$ driving as still driving below the speed limit and, at worst, cutting the safety distance. In those cases, it is rarely the case that the driver is punished in reality.

$$p_i(AS_{-i}, S_i) = a_i \left( \frac{N-1}{N} AS_{-i} + \frac{1}{N} S_i \right)$$

The above game depends on the following parameters: the speed of each strategy ($S_F$, $S_S$, and $S_A$), and the idiosyncratic factor in accident probabilities ($a_F$, $a_S$, and $a_A$). In the experiment below, we focus on the following specific choice:

$$S_F = 2, S_S = 1, S_A = 0.5; a_F = 0.35, a_S = 0.3, a_A = 0, N = 10.$$

## Experimental treatments

The game proposed in the previous section will serve as our control condition for the experiment ("*Control*"). The main interest of the paper is to test the effectiveness of different policy conditions in terms of reducing the proportion of $F$ players and the average speed of the population ($AS$). The reason for the intervention is that $F$ players create a negative externality, and there would be too many of them in equilibrium with respect to the social optimum. But the game is not strictly speaking a social dilemma, in the sense that there are (low) values of risk aversion, and (low) frequencies of $F$ for which an increase in the frequency of $F$ players increases social welfare (that is, the payoff of those extra $F$ players compensates the reduction of payoffs of others). In this section, we derive results suggesting that behavior may be affected by different types of punishment (*Exogenous* and *Endogenous*) as well as the framing of the environment (*Framing*). In addition, it is plausible that different people may react differently to punishment or have different beliefs, and indeed we will show that outcomes may vary for different groups of the population.

*Exogenous* **(punishment).** The experimenter imposes probabilistic fines for players choosing $F$. This only affects participants who were not in an accident, since those in an accident already lose their whole experimental endowment. This policy imposes a penalty for choosing action $F$ which takes effect with some probability $p$, which has been shown to impact real-world driving behavior [31, 36]. Denote the penalty amount to be $C$ and the probability it is imposed to be $q$.

*Endogenous* **(punishment).** In this condition, players can, at a personal cost, impose a penalty $C$, to users of $F$ strategy. The probability $p$ of the punishment is the same as in *Exogenous* but its size depends on how many group members decide to punish the $F$ choosing players. Social sanctioning has been shown to support mutual cooperation in large groups [12,

37–39]), so we assume that there will be a group of individuals (at frequency $p$ in the population) who get some positive utility from punishing individual who flout a social norm. In a driving context the punishment can be to undertake (dangerous) actions, such as retaliatory driving [40] against social norm violators.

Finally, in *Framing* the "punishment" would be simply psychological, the players are primed to think that using strategy $F$ violates a social norm, which would yield a disutility that we can also denote $C$ (in this case we can assume the chance of getting the disutility is $p = 1$). Such social sanctions have been shown to influence behavior in lab settings [41] as well as in real-world driving environments [33, 34].

For these treatments we propose the following the following hypothesis:

**Hypothesis 1** *The proportion of players choosing F will be lower in Exogenous, Endogenous and Framing than in Control.*

## Experimental design

**Participants and sessions.** Experiments were conducted between January and May of 2018 at University College London's Experimental Laboratory for Finance and Economics. The experiment and the corresponding protocol was approved by the University College London Review Board. Participants signed an informed consent form before accessing the experiment. Each participant interacted in one policy condition. We conducted 8 sessions for each condition for a total of 32 experimental sessions. Each session consisted of between 8 and 12 participants. Each experiment session lasted approximately 90 minutes and none of them lasted longer than 2 hours. The average payment was 23.14 GBP. The maximum payment was 43.9 and the minimum was 5.1. These numbers include a 5GBP showup fee. At the end of each session participants provided demographic information about gender, risk preference, age, and experience with driving. The S1 File presents the data for all 326 participants and checks to ensure that our conditions were balanced across the demographic variables.

**Task.** After instructions and a test of comprehension, participants interacted in a multi-round decision-task. In order to avoid strange behavior associated with the final round of the session, the number of rounds was randomly determined to be between 17 and 25 and the participants did not know which round would be the final one in their session. Starting in round 18, there was a $\frac{2}{3}$ chance that another round would be played. This process continued until round 25 was reached, which was determined to be the last round. Participants were told that "The experiment will last between 18 and 25 rounds. The exact number of rounds is randomly determined by the computer." A computer error stopped one session in round 17 instead of round 18. In purity, it is not true that subjects could not know which one was the last round. Any subjects who reached round 25 did. However, this situation never happened. In each round, participants made two incentivized choices: (1) a driving style choice and (2) a guess about the driving style choices of other participants in the room. The remainder of this subsection describes the choice environment that is the same across policy conditions. The exact experimental instructions for all conditions are in the S1 File, including screen shots.

In each round, every participant chose whether to drive "Fast", "Slow", or "Auto". The payoffs for each choice were consistent with the parametrization described in the previous section. Because participants were paid for one randomly selected round, the payoffs were scaled (by 14). In this way, payoffs were represented as GBP during the task. Thus, conditional on not being in an accident in a given round, the participants who chose Fast, Slow, and Auto earned £28, £14, and £7, respectively. In addition, the probabilities of being in an accident were $a_F = 0.35$, $a_S = 0.3$, $a_A = 0$ times the average speed, $AS$. The choice of just three strategies was done

because it preserved a little of the richness of real driving choices, which are really in a continuum, but it allowed for sufficiently stark analysis of choices. With a continuum, we would have had to examine a shift in a distribution between treatments with relatively little data.

In each round, every participant submitted their beliefs about the proportion of participants in the room who would choose Fast, Slow, and Auto. Having just three actions has the advantage of facilitating this belief elicitation. They did so by using the computerized "triangle tool" which allowed participants to make their guess by dragging a point within a triangle where each vertex of the triangle represents a guess where 100% of the participants in the room are choosing one driving style. The amount a participant earned from their guess was £5 minus the difference between their guessed distribution of driving styles and the actual distribution of driving styles in that round. A perfect guess earned £5 and a very inaccurate guess earned £0. A participant earns £5 minus .05 times the number of percentage points their guess is away from the actual population. Consider an example where a participant's guess of Fast, Slow, and Auto players are 50%, 10%, and 40%, respectively, where the actual percentage of Fast, Slow, and Auto players are 50%, 35%, and 15%, respectively. In this example, the participant's Fast guess is 0 percentage points away from the actual population, the Slow guess is 25 percentage points away from the actual population, and the Auto guess is 25 percentage points away from the actual population. In total, the participant's guess deviates from the actual population by 50 percentage points (0+25+25) which would earn that participant £2.50 (£5—.05 x 50). If a participant's guess deviates from the actual population by 100 or more percentage points, the participant earns £0 for their guess.

The triangle tool was also used by participants to calculate the probability of an accident for each driving style conditional on a possible distribution of driving styles. The probability of being in an accident (and earning £0) for each driving style was updated when the participant changed their guess about the population. This way, participants could compare the probabilities of accidents for different driving style choices when facing different beliefs about the distribution of players in the population.

Starting in round 2, participants had complete information about their choices, the choices of other participants in the room, and their payoffs in all previous rounds. In addition, a picture was shown in the top-left of the screen which showed the distribution of driving style choices in the previous round as well as that participant's guess about the distribution in the previous round.

After every participant submitted their driving choice and their guess about the distribution of the other participants in the room, they were shown a results screen summarizing the past round. This screen showed the participant's earnings based on the accuracy of their guess about the population. In addition, each participant was informed about their probability of being in an accident, the realization of this event, as well as their total payoff from their driving choice.

In addition, after all of the driving choice rounds, participants were asked their gender and were tasked with making incentivized decisions in a multiple-price list to elicit risk preferences [42]. Finally, once the driving rounds were over, we probed into the role of social norms by asking the participants about their empirical and normative expectations [43]. This short questionnaire is discussed in detail below.

**Policy conditions.** In accordance with the treatments proposed above, in our experimental design we implemented the following policy conditions:

1. *Control*. Participants interacted in the experiment described above. Participants chose between driving "Fast", "Slow", or "Auto" and were incentivized to guess the distribution of these driving types within the "population" of other participants.

2. *Framing*. Participants chose between driving "Fast", "Slow", or "Safe" and were incentivized to guess the distribution of these driving types within the "community" of other participants. This type of associative framing has been observed to increase contribution rates in public goods games [41].

3. *Exogenous* (punishment). Participants who chose Fast were fined with $C = £4$ with a 25% chance each round. This fine only applied to participants who were not in an accident in that round.

4. *Endogenous* (punishment). Participants who chose Fast had a 25% chance to pay a fine of $C = £X$. $X$ was determined every round in the following way. When participants were making a driving style choice and their guess about the population, they also had to choose whether to contribute £1 into a fund used to punish $F$ players. The fine amount ($X$) equaled the number of participants who contributed to the punishment fund times 2.5. This fine only applied to participants who were not in an accident in that round.

## Results

### Average speed

In a session, the individual driving choices (of Fast, Slow, or Auto) determine the population's Average Speed (AS). AS is an important measure because it determines the probability of an accident for the Fast and Slow players. In this way, AS is a general measure of the overall safety of a driving environment. Figs 1 and 2 plot the AS separated by condition and also by gender. When analyzing all participants (Fig 1), it is clear that none of the policy conditions produce

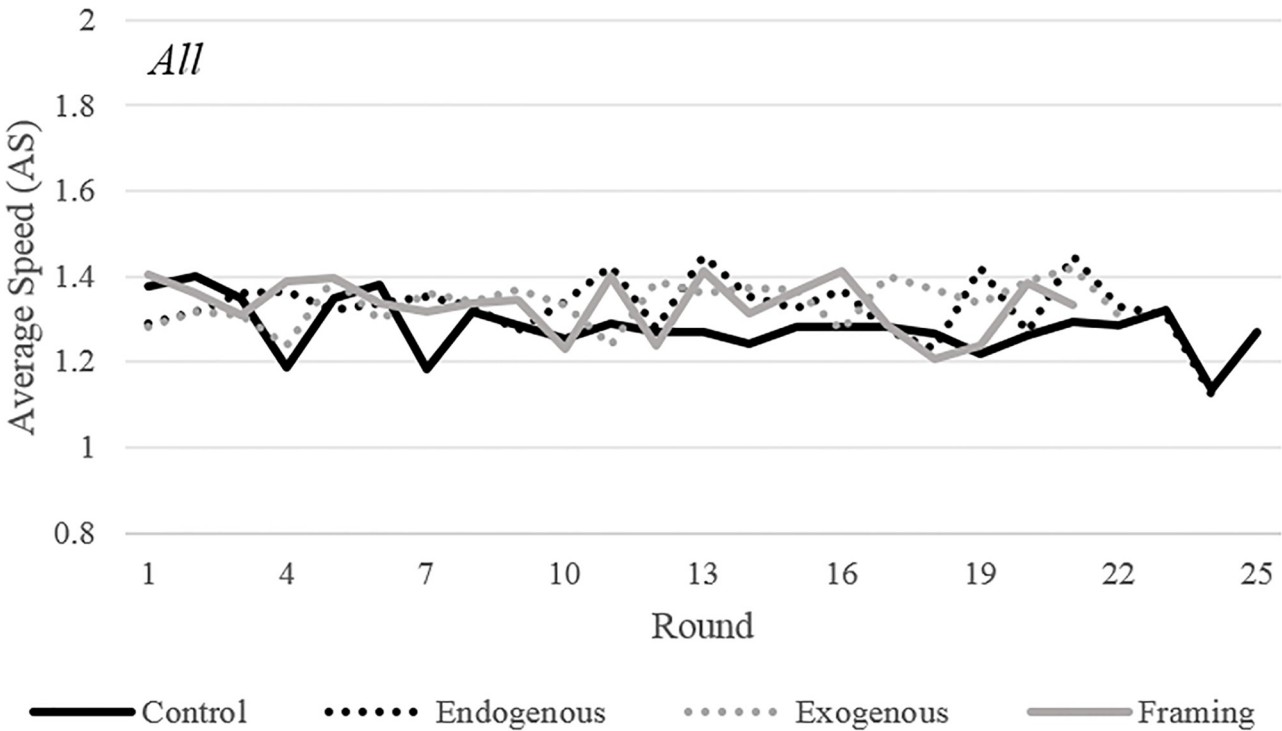

**Fig 1. Average speed by condition.** Average speed by condition. The average speed is 2 when all participants choose Fast and 0.5 when all participants choose Auto. Each line represents the average of data from 8 sessions.

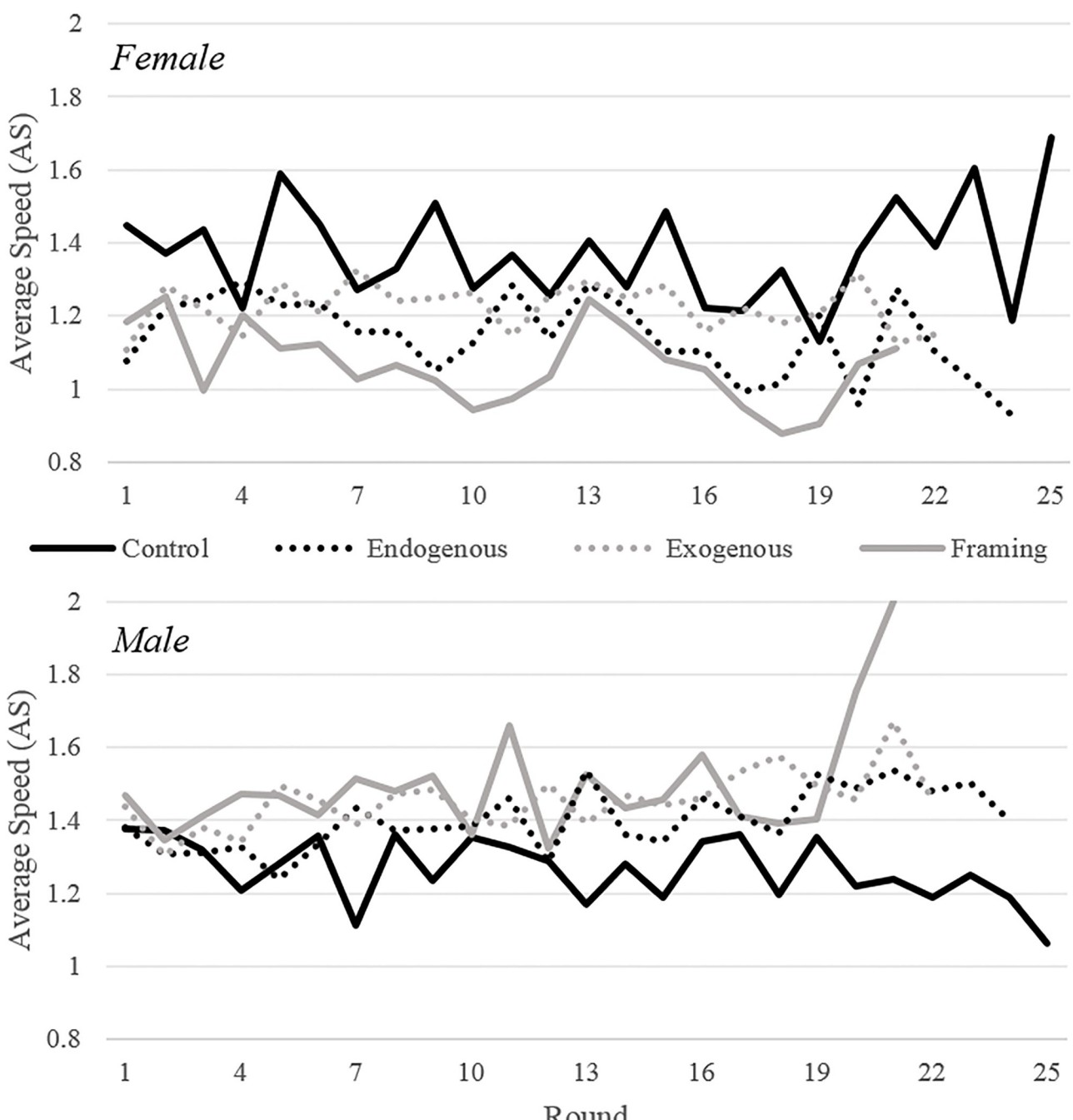

**Fig 2. Average speed by condition and gender.** Average speed by condition and gender (top, female; bottom, male). The average speed is 2 when all participants choose Fast and 0.5 when all participants choose Auto. Each line represents the average of data from 8 sessions.

systematically lower AS than the AS observed in *Control*. However, all three policies have a striking effect when analyzing male and female participants separately (Fig 2). For female participants, the AS in all three conditions is lower than *Control*, while for male participants, the AS in all three conditions is higher than *Control*. Note that the behavior of females is in accordance with the theoretical predictions, but the one of males goes directly against the hypotheses we formulated.

**Table 1. Average Speed by condition and gender.**

|  | *Control* | *Endogenous* | *Exogenous* | *Framing* | AnyPolicy |
|---|---|---|---|---|---|
| All participants | 1.29 | 1.34 | 1.34 | 1.33 | 1.34 |
| Male participants | 1.28 | 1.40 | 1.44 | 1.45 | 1.43 |
| Female participants | 1.35 | 1.15 | 1.23 | 1.07 | 1.15 |
| Male—Female | -0.07 | 0.25 | 0.22 | 0.38 | 0.28 |
| Diff (p-value) | .413 | .006 | .090 | .018 | <.001 |

While 18 (out of 25) rounds in *Control* (top-left panel) show female participants with higher AS than male participants, neither gender chooses systematically higher AS. In the policy conditions, a strikingly different pattern emerges. For any round within any of the 3 policy conditions, it is always the case that the average AS of male participants is higher than the average AS of female participants. To further explore this finding, we calculate the AS within each session averaged across all rounds (this yields one number for each session providing 8 numbers in a condition). In addition, we analyze the AS realizations pooling data from all three policy conditions ("AnyPolicy"; which has 24 realizations). Table 1 shows the average of these AS realizations separated by condition and gender along with *p*-values from two-sample *t*-tests.

In *Control*, there is no difference in the AS across gender. The AS of female participants is significantly lower than the AS of male participants in *Endogenous* and *Framing* ($p = 0.006$ and $p = 0.018$, respectively). For *Exogenous* the difference is not significant, but it has the correct sign, and the *p*-value is 0.090. Note, though, that the subject pool is slightly more risk-averse on average (although the difference only has a *p*-value of 0.06). This could perhaps partially explain the non-significant results in *Exogenous*. We can therefore state the first result of our analysis:

**Result 1.** In *Control*, the AS does not differ by gender. In each policy condition, the AS of female participants is lower than the AS of male participants. This is a particularly strong effect in *Endogenous* and *Framing*.

## Driving choices

Seeing that policies have different effects on the AS (and earnings, cf. S1 File) of male and female participants, we now focus on their driving choices, specifically, to further disentangle the effect of *Endogenous*, *Exogenous*, and *Framing*. Table 2 shows the percentage of driving choices observed across all rounds separated by condition and gender.

When analyzing all participants (left panel), the profile of driving choices in *Framing* and *Exogenous* are not significantly different from *Control* (Pearson's Chi-Squared $p = 0.665$ and 0.144, respectively). *Endogenous* shows the largest effect with a profile of driving choices that is significantly different from *Control* at the 0.016 level. These results do not control for the fact

**Table 2. Driving choices by condition and gender.**

| | *All* | | | *Female* | | | *Male* | | |
|---|---|---|---|---|---|---|---|---|---|
| Condition | % Fast | % Slow | % Auto | % Fast | % Slow | % Auto | % Fast | % Slow | % Auto |
| *Control* | 46.7 | 19.7 | 33.6 | 49.0 | 21.8 | 29.1 | 44.3 | 17.5 | 38.2 |
| *Framing* | 46.0 | 21.0 | 33.0 | 33.8 | 23.3 | 42.9 | 56.9 | 19.0 | 24.2 |
| *Exogenous* | 48.7 | 20.8 | 30.5 | 41.4 | 22.3 | 36.3 | 58.2 | 18.8 | 23.0 |
| *Endogenous* | 44.5 | 23.7 | 31.8 | 34.7 | 30.8 | 34.6 | 56.2 | 15.2 | 28.6 |
| AnyPolicy | 46.4 | 21.9 | 31.7 | 36.9 | 25.8 | 37.4 | 57.1 | 17.8 | 25.4 |

that three hypotheses are tested: *Control*-vs-*Framing*, *Control*-vs-*Exogenous*, and *Control*-vs-*Endogenous*. A conservative way to correct for this is to apply a Bonferroni correction which assumes independence across each test. In doing so, *Endogenous* has a profile of driving choices that is significantly different from *Control* at the 0.048 level. The pooled AnyPolicy condition is also significantly different from *Control* at the 0.001 level. As can be seen, the impact of each policy is (at best) rather small. However, as with AS (Result 1), each condition has a large effect on driving choices within gender. The center panel of Table 2 shows that, compared to *Control*, each policy has fewer female participants who choose Fast and more female participants who choose Auto. This systematic effect go in opposite direction for male participants. As shown in the right panel of Table 2, compared to *Control*, each policy has more male participants who choose Fast and less male participants who choose Auto. All 6 comparisons are significantly different from their respective *Control* at the $p < 0.001$ level (Pearson's Chi-Squared). When applying a Bonferroni correction for 6 tests, 5 out of the 6 tests are still significant at the *ol*) is significant at the $p = 0.008$ level. Comparing AnyPolicy with *Control* also shows a significant difference at the $p < 0.001$ level. As previously remarked, this asymmetry between the genders is both striking and rather unexpected, particularly as regard the behavior of males.

We can further explore this relationship while controlling for independent variables. Using the variables described above, we define **X** as the vector of 2 participant-specific variables (Female and Risk) and 4 round-specific variables (P.Fast, P.Slow, P.Auto, and Late) for which we control. In addition, we define **Z** as a vector containing all possible interactions between the model's condition variables and **X**. For example, when using the AnyPolicy variable, as in model (1), **Z** contains 6 interaction terms (AnyPolicy*Female, AnyPolicy*Risk, AnyPolicy*P. Fast, AnyPolicy*P.Slow, AnyPolicy*P.Auto, and AnyPolicy*Late). In other models, such as model (2), **Z** contains 18 interaction variables.

We use this set of independent variables to explain the dependent variable of driving choice (which is either Fast, Slow, or Auto). We address the following question:

**Question 1:** If the presence of a policy deters Fast players, then which non-Fast action is chosen by these deterred players? And which non-Fast action is chosen by these deterred players in each policy?

We employ a multinomial logistic regression which, for each model, conducts two independent binary logistic regressions in which the Fast driving choice is used as a reference for which the other Slow and Auto are regressed against. As shown below, model (1) uses the pooled "AnyPolicy" independent variable:

$$\ln\left(\frac{p(\text{Slow})}{p(\text{Fast})}\right) = \text{AnyPolicy} \cdot \beta_{1,S} + \mathbf{X}\beta_{\mathbf{X},S} + \mathbf{Z}\beta_{\mathbf{Z},S} + \beta_{0,S}$$

$$\ln\left(\frac{p(\text{Auto})}{p(\text{Fast})}\right) = \text{AnyPolicy} \cdot \beta_{1,A} + \mathbf{X}\beta_{\mathbf{X},A} + \mathbf{Z}\beta_{\mathbf{Z},A} + \beta_{0,A}$$

On the other hand, model (2) separately identifies each policy condition:

$$\ln\left(\frac{p(\text{Slow})}{p(\text{Fast})}\right) = \text{Endogenous} \cdot \beta_{1,S} + \text{Exogenous} \cdot \beta_{2,S} + \text{Framing} \cdot \beta_{3,S} + \mathbf{X}\beta_{\mathbf{X},S} + \mathbf{Z}\beta_{\mathbf{Z},S} + \beta_{0,S}$$

$$\ln\left(\frac{p(\text{Auto})}{p(\text{Fast})}\right) = \text{Endogenous} \cdot \beta_{1,A} + \text{Exogenous} \cdot \beta_{2,A} + \text{Framing} \cdot \beta_{3,A} + \mathbf{X}\beta_{\mathbf{X},A} + \mathbf{Z}\beta_{\mathbf{Z},A} + \beta_{0,A}$$

**Table 3. Driving choice (relative to Fast).**

|  | Slow (1a) | Auto (1b) | Slow (2a) | Auto (2b) |
|---|---|---|---|---|
| AnyPolicy | 0.494 | 0.664 | - | - |
|  | (0.79) | (1.16) |  |  |
| AnyPolicy*Female | 0.456[a] | 0.770** | - | - |
|  | (1.82) | (3.02) |  |  |
| *Endogenous* | - | - | 0.853 | 0.876 |
|  |  |  | (1.20) | (1.53) |
| *Endogenous*\*Female | - | - | 0.744* | 0.732* |
|  |  |  | (2.36) | (2.41) |
| *Exogenous* | - | - | 0.695 | 1.166[a] |
|  |  |  | (0.92) | (1.90) |
| *Exogenous*\*Female | - | - | 0.192 | 0.599[a] |
|  |  |  | (0.59) | (1.66) |
| *Framing* | - | - | -0.015 | 0.036 |
|  |  |  | (-0.02) | (0.05) |
| *Framing*\*Female | - | - | 0.375 | 0.997* |
|  |  |  | (1.37) | (2.47) |
| Female | 0.139 | -0.200 | 0.180 | -0.204 |
|  | (0.66) | (-1.05) | (0.90) | (-1.06) |
| Risk | 0.141 | -0.137 | 0.146 | -0.137 |
|  | (0.95) | (-1.58) | (1.02) | (-1.57) |
| P.Fast | -1.261*** | -0.539* | -1.261*** | -0.539* |
|  | (-6.04) | (-2.26) | (-6.05) | (-2.26) |
| P.Slow | 0.215* | 1.098*** | 0.215* | 1.100*** |
|  | (2.01) | (3.81) | (2.01) | (3.80) |
| P.Auto | 0.217 | 1.964*** | 0.217 | 1.966*** |
|  | (1.18) | (6.69) | (1.18) | (6.69) |
| Late | -0.088 | 0.069* | -0.087 | 0.069* |
|  | (-0.94) | (2.19) | (-0.93) | (2.17) |
| {*Condition*}\*{Risk, P.Fast, P.Slow, P.Auto, Late}, Session Size, & PropFemale | ✓ | ✓ | ✓ | ✓ |
| *N* | 6749 | 6749 | 6749 | 6749 |
| Pseudo-$R^2$ | 0.163 | | 0.166 | |

*t* statistics in parentheses

[a] $p < 0.10$,

\* $p < 0.05$,

\*\* $p < 0.01$,

\*\*\* $p < 0.001$

Table 3 presents the maximum likelihood estimates for the relevant variables. Standard errors are clustered at the session level. We use the 'mlogit' function with the "vce" option in Stata. Since observations are independent across sessions (but not within sessions), errors are clustered at the session level. Participants-level fixed effects are not included because each participant experiences only one condition. Tables 8 and 9, and report on multinomial logit models using the same Stata options (see S1 File for the estimates of all variables in models (1) and (2)). In addition, we find similar results to that shown in Table 3 in a model using a binary logistic regression where the dependent variable is Fast (1) or either of the non-Fast options (0). Estimations of this logit model are in included in S1 File. Furthermore, a previous version

of this manuscript finds similar results when controlling for earnings and experienced accidents in previous rounds. Columns (1) and (2) show the estimates of models (1) and (2) which address question (1).

Columns (1a) and (1b) show that female participants are more likely to choose non-Fast driving choices in the presence of a policy. Relative to the Fast driving choice, female participants are more likely than males to choose Slow ($p = 0.069$) or Auto ($p = 0.003$) in the presence of a policy condition. Furthermore, columns (2a) and (2b) show that this responsiveness is mostly present in *Endogenous* and *Framing*. In both *Endogenous* and *Framing*, female participants are more likely to shift from Fast into Auto ($p = 0.016$ and $0.013$, respectively). In *Exogenous*, female participants are also marginally more likely to shift from Fast into Auto ($p = 0.096$). In addition, in *Endogenous*, female participants are likely to shift from Fast into Slow ($p = 0.018$). The magnitude of these effects is displayed in log odds. Compared to male participants in *Control*, female participants in *Endogenous* have a 0.744 increase in the log odds of choosing Slow (relative to Fast) and a 0.732 increase in the log odds of choosing Auto (relative to Fast). Therefore, we can state our next finding:

**Result 2.** In the presence of any policy condition, female participants are more likely to choose Auto (relative to Fast). This is a particularly strong effect in *Endogenous* and *Framing*. In addition, female participants are more likely choose Slow (relative to Fast) in *Endogenous*.

## Empirical expectations

Having assessed the effects of the policies we are considering on average speed and driving choices, we can now move to the analysis of the subjects's empirical expectations in order to begin unveiling the role of social norms in these results. Following [43], a social norm exists in a group if a majority of members share empirical expectations (beliefs about what most others in the group will do) and normative expectations (beliefs about what most others in the group believe one should do). In each round, a participant submits a belief about the proportion of Fast, Slow, and Auto players that will be present in the group. The accuracy of this elicited belief determines the amount of earnings in that round. The average earnings from these belief elicitations separated by condition and gender are shown in Table 4.

When analyzing all participants (top row of Table 4), each policy condition produces significantly higher payoffs relative to *Control* ($p < 0.001$ using a two-sample $t$-test for all 3 pairwise comparisons). This means that participants are more accurate at predicting the driving choices of others in the presence of any policy condition. Furthermore, earnings from belief elicitations are not different across gender in *Control*, *Endogenous*, or *Exogenous*. Female participants are marginally more accurate in *Framing* ($p = 0.066$). This suggests that inaccurate beliefs of female participants cannot explain the observed differences in AS, driving choice payoffs, or driving choices (Results 1 and 2). In fact, female participants are marginally more accurate than male participants at predicting the choices of others.

**Table 4. Earnings from belief elicitation by condition and gender (£).**

|  | *Control* | *Endogenous* | *Exogenous* | *Framing* | **AnyPolicy** |
|---|---|---|---|---|---|
| All participants | 3.05 | 3.24 | 3.39 | 3.18 | 3.28 |
| Male participants | 3.03 | 3.23 | 3.37 | 3.13 | 3.24 |
| Female participants | 3.08 | 3.25 | 3.41 | 3.23 | 3.30 |
| Male—Female | -0.05 | -0.02 | -0.04 | -0.10 | -0.06 |
| Diff (p-value) | .362 | .620 | .449 | .066 | 0.034 |

As with driving choices, we use the previously described set of independent variables (condition type, **X**, and **Z**) to explore the relationship between beliefs and policies. We address the following question.

**Question 2** Does the presence of a policy change beliefs about Fast/Slow/Auto players in the population? And which specific policy changes the beliefs about Fast/Slow/Auto players in the population?

For models (3) and (4), we employ three separate linear regressions to address each question. The dependent variable in each model is the proportion of a certain type of player the participant guesses will be in the population (as elicited using the triangle tool described above). Model (3) uses the pooled "AnyPolicy" independent variable whereas model (4) separately identifies each policy condition. Table 5 presents the maximum likelihood estimates for the relevant variables. Standard errors are clustered at the session level. As with the analysis on driving choices, the column number aligns with the model number and question number.

Column (3c) shows that female participants facing a policy condition believe that the population will consist of more Auto players ($p = 0.028$). Furthermore, column (4c) shows that this effect is mostly present in *Endogenous* ($p = 0.005$) and marginally present in *Framing* ($p = 0.096$).

Given that female participants believe Auto will be chosen more often in the presence of a policy, what driving choice do they believe will be chosen less often? Interestingly, this depends on the specific policy. Columns (4a) and (4b) show that female participants in the *Endogenous* treatment believe there will be less Fast players ($p = 0.023$), whereas in the *Framing* treatment they believe there will be less Slow players ($p = 0.028$).

**Result 3.** In the presence of any policy, particularly *Endogenous*, female participants believe others are more likely to choose Auto. In addition, female participants believe others are less likely to choose Fast in *Endogenous* whereas, in *Framing*, female participants believe others are less likely to choose Slow.

Thus, we have seen that female participants in *Control*, when compared to their male counterparts, believe that others are less likely to select Auto and more likely to select Slow. Risk preference in *Control* does not explain beliefs and interactions with risk-preference are not consistently significant. Finally, participants in *Control* believe that others are less likely to choose Slow and more likely to choose Auto in late rounds.

## Personal normative beliefs and normative expectations

Empirical expectations have already been elicited when we asked about the proportion of participants who would choose each driving style, but we still need to elicit their normative expectations. Accordingly, participants are first asked the following question about their personal normative beliefs.

*Question (1): In general, what do you think another participant in the experiment should do in this situation?*

After submitting their answer, participants are asked a follow-up question that assesses their normative expectations and their accuracy as compared to the general ones of the population. Question (1) is not incentivized. After submitting the answer to Question (1), participants are told that if their answer to Question (2) was the same as the most-chosen answer to Question (1) within their population, they would earn £2.

*Question (2): In general, what do you think others believe another participant in the experiment should have done? (This is the same as asking 'what driving method do you think most people in the room chose to answer Question (1)')*

**Table 5. Beliefs about the driving choices in the population.**

| | Fast Belief (3a) | Slow Belief (3b) | Auto Belief (3c) | Fast Belief (4a) | Slow Belief (4b) | Auto Belief (4c) |
|---|---|---|---|---|---|---|
| AnyPolicy | -0.564 | -3.266 | 3.830 | - | - | - |
| | (-0.17) | (-1.01) | (0.98) | | | |
| AnyPolicy*Female | -2.697 | -1.816 | 4.513* | - | - | - |
| | (-1.43) | (-1.28) | (2.31) | | | |
| *Endogenous* | - | - | - | -2.022 | -0.356 | 2.377 |
| | | | | (-0.51) | (-0.12) | (0.54) |
| *Endogenous*\*Female | - | - | - | -4.636* | -1.185 | 5.821** |
| | | | | (-2.39) | (-0.80) | (3.00) |
| *Exogenous* | - | - | - | 5.408 | -9.635* | 4.227 |
| | | | | (1.29) | (-2.42) | (0.75) |
| *Exogenous*\*Female | - | - | - | -3.884 | -0.879 | 4.763 |
| | | | | (-1.54) | (-0.42) | (1.50) |
| *Framing* | - | - | - | -2.326 | -1.327 | 3.653 |
| | | | | (-0.61) | (-0.29) | (0.82) |
| *Framing*\*Female | - | - | - | 0.261 | -3.163* | 2.902[a] |
| | | | | (0.16) | (-2.31) | (1.72) |
| Female | 1.053 | 2.256* | -3.309* | 0.972 | 2.394* | -3.366* |
| | (0.68) | (2.17) | (-2.15) | (0.63) | (2.48) | (-2.25) |
| Risk | 0.006 | -0.397 | 0.391 | 0.009 | -0.400 | 0.391 |
| | (0.01) | (-0.92) | (0.81) | (0.02) | (-0.92) | (0.81) |
| P.Fast | 6.688** | -10.180*** | 3.492 | 6.677** | -10.163*** | 3.486 |
| | (3.16) | (-4.33) | (1.11) | (3.16) | (-4.31) | (1.10) |
| P.Slow | 0.762 | -4.037* | 3.274 | 0.724 | -3.976* | 3.252 |
| | (0.24) | (-2.08) | (1.01) | (0.23) | (-2.05) | (1.00) |
| P.Auto | 2.661 | -10.380*** | 7.720* | 2.705 | -10.451*** | 7.746* |
| | (1.29) | (-4.22) | (2.43) | (1.31) | (-4.24) | (2.43) |
| Late | 0.420 | -4.223*** | 3.803* | 0.405 | -4.200*** | 3.795* |
| | (0.29) | (-9.88) | (2.64) | (0.28) | (-9.53) | (2.63) |
| {*Condition*}*{Risk, P.Fast, P.Slow, P.Auto, Late}, Session Size, & PropFemale | ✓ | ✓ | ✓ | ✓ | ✓ | ✓ |
| *N* | 6749 | 6749 | 6749 | 6749 | 6749 | 6749 |
| Adj.-$R^2$ | 0.057 | 0.119 | 0.086 | 0.080 | 0.138 | 0.091 |

*t* statistics in parentheses

[a] $p < 0.10$,

\* $p < 0.05$,

\*\* $p < 0.01$,

\*\*\* $p < 0.001$

In this respect, one possible concern about our design is that we elicit beliefs after participants are treated, and hence the beliefs could be endogenous to that treatment, as has been observed in some cases, because of self-serving biases [44, 45]. However, if elicitation occurs before the behavior, people may focus their attention on the norms that prevail in that situation, and may thus affect behavior, and in doing so they incur in what is called situational cues biases [1, 46–49]. After considering these possibilities, we chose to elicit beliefs after the behavioral choice as most appropriate for our purposes. Also, our main interest is in showing the impact of treatments in the choice of actions. By placing belief elicitation before treatments,

**Table 6. Personal normative beliefs by condition.**

| Condition | % Fast | | | | % Auto | | | |
|---|---|---|---|---|---|---|---|---|
| | Male | Female | Male-Female | Diff (p-value) | Male | Female | Male-Female | Diff (p-value) |
| *Control* | 30.8 | 34.1 | -3.3 | 0.933 | 53.8 | 41.5 | 12.3 | 0.376 |
| *Framing* | 48.8 | 30.6 | 18.2 | 0.163 | 29.3 | 50.0 | -20.7 | 0.104 |
| *Exogenous* | 47.2 | 41.7 | 5.5 | 0.775 | 30.6 | 41.7 | -11.1 | 0.415 |
| *Endogenous* | 51.3 | 23.9 | 27.4 | 0.017 | 33.3 | 45.7 | -12.4 | 0.351 |
| AnyPolicy | 49.1 | 32.3 | 16.8 | 0.011 | 31.0 | 45.4 | -14.4 | 0.030 |

the treatment would include the beliefs and there would be a confound that we wished to avoid.

The percentage of participants who state Fast or Auto as their personal normative beliefs are shown in Table 6. Slow is omitted because it is the least chosen option and because the percentage of participants who choose Slow are not statistically different across gender.

In *Control*, the percentage of participants who state Fast as their personal normative belief is not different across genders ($p = 0.933$).References to $p$-values in Tables 6, 7, and 10 are from Pearson's Chi-Squared tests. The same is true for stating Auto ($p = 0.376$). In the presence of any policy, compared to their male counterparts, female participants are less likely to state Fast and more likely to state Auto as their personal normative belief ($p = 0.011$ and $p = 0.030$ using a two-sample chi-squared test comparing Fast with non-Fast and Auto with non-Auto). This effect is largely driven by the behavior in *Endogenous* and *Framing*. More specifically, compared to male participants, Fast is chosen significantly less often by female participants in *Endogenous* ($p = 0.017$) and Auto is chosen marginally more often by female participants in *Framing* ($p = 0.104$).

The percentage of participants who state Fast or Auto as their normative expectations are shown in Table 7.

In *Control*, the percentage of participants who state Fast as their normative expectation is not different across genders ($p = 1.000$). The same is true for stating Auto ($p = 0.646$). In the presence of any policy, compared to their male counterparts, female participants are less likely to state Fast and more likely to state Auto as their personal normative belief ($p = 0.044$ and $p = 0.009$ using a two-sample chi-squared test comparing Fast with non-Fast and Auto with non-Auto). This effect is largely driven by behavior in *Endogenous*. More specifically, compared to male participants, Fast is chosen significantly less often by female participants ($p = 0.031$) and Auto is chosen very close to significantly more often by female participants in *Endogenous* ($p = 0.055$).

Both personal normative beliefs and normative expectations consistently differ across gender in the presence of a policy. This is a first important finding to explain the effect of policies, which is particularly surprising with respect to male behavior. Males normative beliefs and

**Table 7. Normative expectations by condition.**

| Condition | % Fast | | | | % Auto | | | |
|---|---|---|---|---|---|---|---|---|
| | Male | Female | Male-Female | Diff (p-value) | Male | Female | Male-Female | Diff (p-value) |
| *Control* | 33.3 | 31.7 | 1.6 | 1.000 | 56.4 | 48.8 | 7.6 | 0.646 |
| *Framing* | 43.9 | 27.8 | 16.1 | 0.219 | 36.6 | 52.8 | -16.2 | 0.231 |
| *Exogenous* | 47.2 | 45.8 | 1.4 | 1.000 | 27.8 | 43.8 | -16.0 | 0.203 |
| *Endogenous* | 43.6 | 19.6 | 24.0 | 0.031 | 33.3 | 54.3 | -21.0 | 0.055 |
| AnyPolicy | 44.8 | 31.5 | 13.3 | 0.044 | 32.8 | 50.0 | -17.2 | 0.009 |

expectations are not moved against Fast because of policies. As with driving choices and belief elicitations, we aim explore the relationship between these social norms and policies. We hence address the following question:

**Question 3:** If specific policies deter the Fast-driving norm, then which non-Fast norm prevails in each policy?

For models (5) and (6), we employ multinomial logistic regressions similar to models (1) and (2). The main difference from those models is that the dependent variable is the norm choice rather than driving choice. In addition, since we only have one data point for each participant, round-specific variables are not included. Tables 8 and 9 present the maximum likelihood estimates of models (5) and (6) for personal normative beliefs and empirical expectations, respectively.

For the personal normative beliefs, column (5b.1) shows that female participants in the presence of a policy are more likely to answer Auto (relative to Fast; $p = 0.057$). Furthermore, column (6b.1) shows that this responsiveness is mostly present in *Endogenous*. In summary, the effect of policies on personal normative beliefs is similar to the effect of policies on driving choices (Result 2). On the other hand, for the normative expectations, we don't observe a statistically significant effect of the presence of a policy. However, as with personal normative beliefs, column (6b.2) shows that female participants in *Endogenous* are more likely to respond Auto (relative to Fast; $p = 0.019$). Column (6b.2) also shows that male participants in *Framing* are less likely to state Auto (relative to Fast; $p = 0.005$). Therefore, we summarize our findings in this regard as follows:

**Result 4.** In the presence of the *Endogenous* policy condition, female participants are more likely to state Auto as their personal normative belief instead of Fast. Male participants are less likely state Auto as their normative expectations instead of Fast in *Framing*.

One interesting question in this context is whether the gender differences in normative beliefs about driving choices extends to the imposition of fines in the *Endogenous* policy condition. We did not ask a question about norms in the case of fines, but the actions do not differ very strongly. Female participants contribute to the punishment fund 13.5% of the time, which is higher than male participants (8.6%). However, these differences are based on only 8 sessions of the *Endogenous* treatment and a logit model which clusters at the session level shows this difference to be insignificant (details in the S1 File).

As stated above, one has a social norm if they have a personal normative belief that is consistent with their normative expectation [43]. In our experiment, such participants will have consistent answers to Question 1 and Question 2. The percentage of such participants are shown in Table 10 divided by gender and the norm.

Male participants are more likely than Female participants to have Fast as their social norm ($p = 0.048$). In addition, Female participants are marginally more likely than male participants to have Auto as their social norm ($p = 0.095$). This leads us to the following finding:

**Result 5.** Female participants are more likely to state Auto as their social norm. Male participants are more likely to state Fast as their social norm.

Results 4 and 5 suggest that female participants create stronger social norms around non-Fast driving choices (especially in *Endogenous* and *Framing*). The combination of results 4 and 5 suggest that the theoretically unexpected behavior of males in the presence of policies is driven by normative beliefs and expectations.

## Discussion and conclusion

In this paper, we have proposed and studied policies to reduce driving speeds, the main factor driving the risk of being injured in vehicle accidents, in the presence of different driving styles.

**Table 8. Personal normative belief (relative to Fast).**

| | Slow (5a.1) | Auto (5b.1) | Slow (6a.1) | Auto (6b.1) |
|---|---|---|---|---|
| AnyPolicy | 0.324 | -0.044 | - | - |
| | (0.19) | (-0.06) | | |
| AnyPolicy*Female | 0.158 | 1.079[a] | - | - |
| | (0.22) | (1.90) | | |
| Endogenous | - | - | 0.400 | 1.449 |
| | | | (0.18) | (1.37) |
| Endogenous*Female | - | - | 1.0413 | 1.493* |
| | | | (1.33) | (2.28) |
| Exogenous | - | - | 0.410 | -0.136 |
| | | | (0.20) | (-0.11) |
| Exogenous*Female | - | - | -0.511 | 0.570 |
| | | | (-0.59) | (0.80) |
| Framing | - | - | 0.306 | -0.793 |
| | | | (0.17) | (-0.88) |
| Framing*Female | - | - | -0.038 | 1.340[a] |
| | | | (-0.05) | (1.87) |
| Female | 0.488 | -0.319 | 0.464 | -0.312 |
| | (0.70) | (-0.66) | (0.67) | (-0.65) |
| Risk | 0.084 | -0.122 | 0.081 | -0.122 |
| | (0.27) | (-1.22) | (0.25) | (-1.23) |
| AnyPolicy*Risk | -0.116 | -0.223 | - | - |
| | (-0.34) | (-1.36) | | |
| Endogenous*Risk | - | - | -0.202 | -0.642** |
| | | | (-0.46) | (-3.23) |
| Exogenous*Risk | - | - | -0.098 | -0.152 |
| | | | (-0.27) | (0.58) |
| Framing*Risk | - | - | -0.093 | -0.063 |
| | | | (-0.25) | (-0.37) |
| Session Size | 0.071 | 0.034 | 0.065 | 0.013 |
| | (0.45) | (0.40) | (0.38) | (0.15) |
| PropFemale | -1.495 | -0.400 | -1.272 | -0.324 |
| | (-1.21) | (-0.54) | (-0.89) | (-0.39) |
| Constant | -1.100 | 0.861 | -1.115 | 1.038 |
| | (-0.50) | (0.95) | (-0.49) | (1.11) |
| N | 326 | 326 | 326 | 326 |
| Pseudo-$R^2$ | 0.043 | | 0.059 | |

$t$ statistics in parentheses

[a] $p < 0.10$,

* $p < 0.05$,

** $p < 0.01$,

*** $p < 0.001$

Specifically, we have considered framing the situation in a safety-conscious manner, fines by an external agent (traffic police), and fines imposed through the implication of the drivers' community. A first conclusion of our research is that neither of these proposals leads to a reduction in average speed. This points out the difficulty of designing mechanisms that lead to

**Table 9. Normative expectations (relative to Fast).**

| | Slow (5a.2) | Auto (5b.2) | Slow (6a.2) | Auto (6b.2) |
|---|---|---|---|---|
| AnyPolicy | 0.553 | -1.010 | - | - |
| | (0.40) | (-1.30) | | |
| AnyPolicy*Female | -0.590 | 0.868 | - | - |
| | (-0.99) | (1.43) | | |
| *Endogenous* | - | - | 0.474 | 0.745 |
| | | | (0.24) | (0.73) |
| *Endogenous*\*Female | - | - | 0.102 | 1.609* |
| | | | (0.15) | (2.35) |
| *Exogenous* | - | - | 0.440 | -0.920 |
| | | | (0.16) | (-0.89) |
| *Exogenous*\*Female | - | - | -1.492 | 0.452 |
| | | | (-1.51) | (0.57) |
| *Framing* | - | - | 0.504 | -2.178** |
| | | | (0.37) | (-2.80) |
| *Framing*\*Female | - | - | -0.305 | 0.970 |
| | | | (-0.43) | (1.43) |
| Female | 0.939[a] | -0.226 | 0.908[a] | -0.251 |
| | (1.94) | (-0.41) | (1.89) | (-0.45) |
| Risk | 0.162 | -0.223* | 0.136 | -0.217[a] |
| | (0.56) | (-1.96) | (0.46) | (-1.89) |
| AnyPolicy*Risk | 0.006 | 0.037 | - | - |
| | (0.02) | (0.25) | | |
| *Endogenous*\*Risk | - | - | 0.048 | -0.470* |
| | | | (0.11) | (-2.16) |
| *Exogenous*\*Risk | - | - | 0.030 | -0.019 |
| | | | (0.06) | (-0.12) |
| *Framing*\*Risk | - | - | -0.017 | 0.380** |
| | | | (-0.05) | (2.66) |
| Session Size | -0.035 | 0.115 | -0.057 | 0.108 |
| | (-0.20) | (1.01) | (-0.28) | (1.03) |
| PropFemale | -1.447 | 0.718 | -1.108 | 0.955 |
| | (-1.11) | (0.58) | (-0.72) | (0.82) |
| Constant | -0.978 | -0.077 | -0.885 | -0.095 |
| | (-0.46) | (-0.06) | (-0.39) | (-0.08) |
| *N* | 326 | 326 | 326 | 326 |
| Pseudo-$R^2$ | 0.048 | | 0.084 | |

*t* statistics in parentheses

[a] $p < 0.10$,

* $p < 0.05$,

** $p < 0.01$,

*** $p < 0.001$

safer driving, or more generally reducing socially harmful behavior. A second, and possibly more important, insight is that policies have heterogeneous effects in the population. In particular the differential effect of policies by gender is markedly different and even counter-intuitive, especially with regards to males. Below we discuss this issue in more detail as its

**Table 10. Social norm by gender (%).**

|  | Auto | Slow | Fast | No norm |
|---|---|---|---|---|
| Male participants | 27.1 | 12.9 | 34.2 | 25.8 |
| Female participants | 38.0 | 12.9 | 25.1 | 24.0 |
| Male—Female | -10.9 | 0.0 | 9.1 | 1.8 |
| Diff (p-value) | 0.048 | 1.000 | 0.095 | 0.800 |

implications may carry over to other mechanisms intended to promote safer or prosocial behavior.

Our findings point to social norms as the reason underlying the differences in choices, although the precise mechanism through which they work is more difficult to assess as different interpretations of our results are possible: indeed, it may be the case that female participants understand that choosing Fast deviates from the social norm, which implicitly amounts to implying that males' understanding of the norms differs from females'. On the other hand, after the norm has been enforced, males that have been punished may have a lower preference for norm-compliance (independent of their belief about the norm). Therefore, the effect of social norms may be mediated by norm-perception, by norm-compliance, or by both, and we cannot make a clear identification of the mediator at work here. More work is needed in this area.

There is some work showing that adherence to social norms may be mediated by gender. Some examples are [50] in the context of study norms, [51] for climate change mitigation behavior, and [52] for adherence to public health measures during the pandemic. One main difference in our case is the context. More important, perhaps, is the fact that in our case males not just comply less, they take advantage of compliers to behave worse. The origin of that gender differential compliance is still underexplained. But there are some hints that the explanation lies in differential brain activity [21]. At the same time, women have different kinds of social networks, and that has also been shown to impact the adherence to fairness norms [53]. An avenue for future research could use our findings and explore them further in this direction.

In our experiments, we have found that policies reduce the average speed of female participants and increase that of male participants. In fact, this is why we do not observe a decreasing of the global average speed. Correspondingly, men increase their earnings at the expense of female participants when policies are implemented. This means that the differences in gender reaction to policies is not without consequences, and female participants are harmed by their more prosocial choices. Interestingly, the choices of female participants are the theoretically predicted ones: in the presence of policy conditions, female participants choose Auto instead of Fast. While this is particularly true in *Endogenous* and *Framing*, we have observed that in *Endogenous* female participants also choose Slow instead of Fast, suggesting that *Endogenous* is the most effective policy among female participants. Our design allows us to suggest possible mechanisms driving these differences in behavior, by using question about personal normative beliefs and empirical and normative expectations to elicit social norms. We thus found that the social norm of male participants is more likely to be Fast than the norm of female participants, which turns out to be Auto. As a result, female participants reduce their use of Fast driving styles, which makes it more profitable for males to use Fast, thereby annulling the effect of the policy. We should note that we obtain the results in spite of the relatively low number of independent observations (which arises because the number of players per session is necessarily large). While the low number of observations is a limitation, the fact that we obtain clear

results is encouraging as it suggests the underlying effect size of our treatments is very large so that we observe it in spite of the large confidence bounds.

In summary, the policies proposed here turn out to be effective at changing female participants' driving behavior, but male participants have an opposing reaction. This is very important for policy. Not only might policies be ineffective, but the behavioral reactions in the population can increase inequity. We already know that behavioral reactions can mitigate the effect of policies (as in the example of seat-belt legislation [54]), but this complete cancellation and worsening of inequality is more worrying. Furthermore, we know that social norms interventions are particularly sensitive to their implementation [55]. Importantly, such policy effects appear to be mediated by social norms, and the difference between social norms of male and female participants is connected with their different response to policies. which are more prevalent among female drivers. Our results suggest that a proper policy to change behavior should appeal directly to people's expectations about what others will do [56], and for the case of reducing driving speed should be preferentially addressed to men. In relation to this, care has to be taken when designing the information to be presented as these type of nudges can actually backfire [55]; cultural factors such as tightness or looseness of the society of interest [57] can also be relevant for such interventions, as the prevailing culture among males is generally looser regarding social norms [58]. Overall, our findings suggest that future research which analyzes driving behavior, or more generally any prosocial behavior, should remain vigilant about the possibility of gender differences in this context mediated by social norms. To give another example, it could be that males find it easier to strategically distort norm-relevant beliefs. As shown in [59], people are able to distort their beliefs about the prevailing social norms if they are likely to be affected by it. These are important aspects to take into account and clarify further in future research.

## Supporting information

**S1 File. Theoretical analysis of the model, complete with proofs of propositions; participant statistics and balance check; tables with full regression results; analysis of earnings; analysis of the contributions to punishment; experimental instructions; screen shots of the experiment software.**
(PDF)

## Author Contributions

**Conceptualization:** Antonio Cabrales, Angel Sánchez.

**Data curation:** Ryan Kendall.

**Formal analysis:** Antonio Cabrales, Angel Sánchez.

**Funding acquisition:** Antonio Cabrales, Angel Sánchez.

**Investigation:** Antonio Cabrales, Ryan Kendall, Angel Sánchez.

**Methodology:** Ryan Kendall.

**Project administration:** Antonio Cabrales.

**Software:** Ryan Kendall.

**Visualization:** Ryan Kendall.

**Writing – original draft:** Antonio Cabrales, Ryan Kendall.

**Writing – review & editing:** Antonio Cabrales, Ryan Kendall, Angel Sánchez.

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
