## [Decision Letter · Decision Letter 0]

3 May 2022

PONE-D-22-09104The effectiveness of prosocial policies: Gender differences arising from social normsPLOS ONE

Dear Dr. Sánchez,

Thank you for submitting your manuscript to PLOS ONE. After careful consideration, we feel that it has merit but does not fully meet PLOS ONE’s publication criteria as it currently stands. Therefore, we invite you to submit a revised version of the manuscript that addresses the points raised during the review process. Please submit your revised manuscript by Jun 17 2022 11:59PM. If you will need more time than this to complete your revisions, please reply to this message or contact the journal office at plosone@plos.org. Please include the following items when submitting your revised manuscript:A rebuttal letter that responds to each point raised by the academic editor and reviewer(s). You should upload this letter as a separate file labeled 'Response to Reviewers'.A marked-up copy of your manuscript that highlights changes made to the original version. You should upload this as a separate file labeled 'Revised Manuscript with Track Changes'.An unmarked version of your revised paper without tracked changes. You should upload this as a separate file labeled 'Manuscript'.

We look forward to receiving your revised manuscript.

Kind regards,

Nikolaos Georgantzis, Dr.

Academic Editor

PLOS ONE

Journal Requirements:

Reviewers' comments:

Reviewer's Responses to Questions

**Comments to the Author**

1. Is the manuscript technically sound, and do the data support the conclusions?

Reviewer #1: Yes

Reviewer #2: Partly

2. Has the statistical analysis been performed appropriately and rigorously? 

Reviewer #1: Yes

Reviewer #2: Yes

3. Have the authors made all data underlying the findings in their manuscript fully available?

Reviewer #1: Yes

Reviewer #2: Yes

4. Is the manuscript presented in an intelligible fashion and written in standard English?

Reviewer #1: Yes

Reviewer #2: Yes

5. Review Comments to the Author

Reviewer #1: Specific remarks:

1.This is an interesting paper that sheds light on how different policies can have differential gender effects. It also adds to the long literature on gender effects. The experiments are well designed (with a small caveta-see below) and the results are clear. However, there are lots of typos and spelling mistakes in the entire document.

2.Recent work has shown that providing information on accidents is distracting and actually increases them due to cognitive fatigue (How safe are safety messages?, Gerald Ullman and Susan Chrysler, Science, 21 Apr 2022, Vol 376, Issue 6591, pp. 347-348, DOI: 10.1126/science.abq1757). Your paper is interesting in this aspect as it can be argued that it shows that ex-ante primes (not while at the task) may be a better solution as it does not add to the cognitive fatigue while driving.

3.Is there work that has looked at differential gender effects on norms? If not, it makes the paper even more relevant.

4.The introduction is not clear and reads confused. It will be a good idea to narrow down what you are doing and writer a clearer introduction.

5.Para 2, line 1. Missing citation. Figure numbers appear as ?? later on in the paper too.

6.Intro para 2, line 5. Do you mean that the context is preferences over electric vs hybrid (which pollute more than electric)? Again, what is the context dependent element here? I found this confusing to read.

7.Intro para 2, line 8. I do not see the continuation of the argument here. You are going from social norms that are context dependent to laboratory experiments. There is no relation. These are two separate arguments.

8.Para 3. There is a substantial information on driving, by age, gender, highways and country roads. It would be good if you put your motivation in that context. A quick google search told me that speeding only accounts for 28% of the accidents. I think what you are doing is more general than that. Your framework can apply to many scenarios. Maybe thinking about other scenarios where this may apply will broaden the appeal of the paper. A discussion to this regard would be useful.

9.Last para, page 2. ….such AS higher deaths from both…

10.I find sentences in the introduction that are unrelated to the previous arguments, line 2, page 3, intro is an example.

11.Page 3, line 8 (is it Manual or Slow?)

12.Page 3, line 9, you mention the strategies without having defined them.

13.You assume that speeding also increases the probability of accidents. This is a ceteris paribus argument. Some clarification on this would be useful.

14.Page-4, line 7, “However” repeats.

15.Page 4, line 11, Can start with a new paragraph. Its easier to read the texts if each para starts with a new argument. Page-4 is a long read. It can have new paras at line 11, 19,

16.Same page, assesS (spelling).

17.Page-5, first para, last line. I found it hard to understand the sentence.

18.In the “Relevant literature” section you can include articles on Transport research. There is a substantial literature on driver behaviors. It would be good to have a summary about what we already know. For example, see, “The impact of road advertising signs on driver behavior and implications for road safety: A critical systematic review,” Transportation Research Part A: Policy and Practice, Volume 122, April 2019, Pages 85-98.

19.Para 2, section 2, is your paper about cooperation?

20.Page 6, Section2, para 2, line 4. Extra space before parenthesis. This repeats in several places.

21.Page 5, beginning, you state that, “Because this literature provides ambiguous messages about the effect of gender, we create a model in Section 3 that does not directly account for gender differences. Instead we allow the data to clarify the direction of effects (Section 5).” Then what is the usefulness of your model?

22.Page 6, para 2: “At this point” is unnecessary. It does not follow from a past argument.

23.Page 7, para 1. Remove “can”.

24.3.1. The game:

a.Explain better how the theoretical models maps into the experimental design. If you want the data to speak for itself, then how relevant is your theoretical framework?

b.“driver” is out of place. Delete.

c.Define variables in pi.

d.The accident probabilities are pi. When defining EU on top of page 8, you state that ai defines accident probabilities?

25.Section 3.2.

a.Please clarify what you mean by imperfectly imposed fines. It becomes clearer later on that they are imposed on those that “are not in an accident.”

b.Endogenous. You have changed two things, the size of the fine which depends on how many vote for it. Better say that it is a variable sign that is increasing in the number of those that vote for the sign. Proably, accounting for beliefs probably controls for this in the analysis. From the results it seems that this should not matter. Discussing this would be useful.

c.Endogenous: “But we social sanctioning…” incomplete sentence.

d.From Proposition 2, can you rank the proportion of players choosing F for each treatment?

e.You define a variable alpha that captures sensitivity to fines. This could capture gender effects too.

26.The results are very interesting. Endogenous and framing nudges work stronger for females. It would be interesting to see a discussion on why this is so. For example, brain research has shown that females are more consensus oriented and “aware” of their immediate surroundings (for example read The Female Brain, Louann Brizendine). These results also add to the literature on gender differences in cooperation and competition where they argue that context matters more for females (see, Gender Differences in Preferences, Rachel Croson and Uri Gneezy, Journal of Economic Literature, Vol. 47, No. 2, June 2009, pp. 448-74.

Reviewer #2: See attached report.........................................................................................................................................................................................................

6. PLOS authors have the option to publish the peer review history of their article (what does this mean?). If published, this will include your full peer review and any attached files.

Reviewer #1: No

Reviewer #2: No

---

## [Author Response · Author response to Decision Letter 0]

28 Jun 2022

Answers to Reviewer 1 and summary of changes made in response to his/her comments

1. This is an interesting paper that sheds light on how different policies can have differential gender effects. It also adds to the long literature on gender effects. The experiments are well designed (with a small caveat-see below) and the results are clear. However, there are lots of typos and spelling mistakes in the entire document.

We thank the Reviewer for his/her praise of our work and we sincerely apologize for the typos and mistakes. In this new version we have tried our best to fix all those issues. 

2. Recent work has shown that providing information on accidents is distracting and actually increases them due to cognitive fatigue (How safe are safety messages?, Gerald Ullman and Susan Chrysler, Science, 21 Apr 2022, Vol 376, Issue 6591, pp. 347-348, DOI: 10.1126/science.abq1757). Your paper is interesting in this aspect as it can be argued that it shows that ex-ante primes (not while at the task) may be a better solution as it does not add to the cognitive fatigue while driving.

We thank the Reviewer for this comment, which seems very interesting to us. Therefore, we have added in the introduction a sentence about this, thanking the Reviewer in a footnote.

3. Is there work that has looked at differential gender effects on norms? If not, it makes the paper even more relevant.

This is indeed an important question which we have addressed in the revised version of the manuscript. We now give some examples of that literature in the review, noting explicitly in the new text “the main difference in our case is the context, on the one hand. More important, perhaps, is the fact that in our case males not just comply less, they take advantage of compliers to behave worse.”

4. The introduction is not clear and reads confused. It will be a good idea to narrow down what you are doing and writer a clearer introduction.

We have significantly reduced and focused the introduction and thereby, we hope, made it clearer.

5. Para 2, line 1. Missing citation. Figure numbers appear as ?? later on in the paper too.

We do not know what is the reason, but we have not been able to nail down this problem because we do not observe any of this in our manuscript. Perhaps it was a product of the compilation of the file in the journal server? In any event, in this new version we are sending a pdf file making sure that there are no missing references.

6. Intro para 2, line 5. Do you mean that the context is preferences over electric vs hybrid (which pollute more than electric)? Again, what is the context dependent element here? I found this confusing to read.

We are sorry this was not clear in the previous version. We now talk about antisocial behaviors, which we now define in the first paragraph, explaining that these are examples of antisocial behaviors that share features with the driving example.

7. Intro para 2, line 8. I do not see the continuation of the argument here. You are going from social norms that are context dependent to laboratory experiments. There is no relation. These are two separate arguments.

We agree with the Reviewer on this point. We now make this clearer by separating in different paragraphs and slightly changing the explanation.

8. Para 3. There is a substantial information on driving, by age, gender, highways and country roads. It would be good if you put your motivation in that context. A quick google search told me that speeding only accounts for 28% of the accidents. I think what you are doing is more general than that. Your framework can apply to many scenarios. Maybe thinking about other scenarios where this may apply will broaden the appeal of the paper. A discussion to this regard would be useful.

We are thankful to the Reviewer for this useful insight. We now mention driver distraction and intoxication as behaviors that cause similar problems and for similar reasons.

9. Last para, page 2. ….such AS higher deaths from both…

We have corrected this. 

10. I find sentences in the introduction that are unrelated to the previous arguments, line 2, page 3, intro is an example.

In order to clarify what we mean here, we have made the explanation about autonomous vehicles a separate paragraph and tried to explain better that they will make antisocial behavior more “profitable” and thus more worrisome.

11. Page 3, line 8 (is it Manual or Slow?)

We now clarify by introducing the words “Non Auto”.

12. Page 3, line 9, you mention the strategies without having defined them.

We are sorry about this. We have taken care of this issue by defining strategies before talking about them.

13. You assume that speeding also increases the probability of accidents. This is a ceteris paribus argument. Some clarification on this would be useful.

We hope that the new explanation about this is clearer and more useful for the reader.

14. Page-4, line 7, “However” repeats.

We have removed the first instance, which indeed was not necessary.

15. Page 4, line 11, Can start with a new paragraph. It’s easier to read the texts if each para starts with a new argument. Page-4 is a long read. It can have new paras at line 11, 19,

We agree with the Reviewer that the previous version was hard to read, so we have included more paragraph breaks. 

16. Same page, assesS (spelling).

Corrected.

17. Page-5, first para, last line. I found it hard to understand the sentence.

We now explain that “Indeed, many previous papers show that people internalize social norms more strongly when they come from the community than when they are imposed by an external authority.”

18. In the “Relevant literature” section you can include articles on Transport research. There is a substantial literature on driver behaviors. It would be good to have a summary about what we already know. For example, see, “The impact of road advertising signs on driver behavior and implications for road safety: A critical systematic review,” Transportation Research Part A: Policy and Practice, Volume 122, April 2019, Pages 85-98.

We thank the Reviewer for pointing out to us this literature. We now discuss it int the following terms: “Given that our experiment has a specific frame we should note there is an extensive literature in transportation science that identifies the effect of gender on driving behavior. Oviedo-Trespalacios et al. (2019) shows that males are distracted more often by roadside signs, but females look at them longer. Jiang (2020) find that women are less aggressive when encountering a yellow light. Interestingly for our research, Jorgensson and Polak (1993) find that although males drive faster in the absence of speeding fines, there is no gender difference when fines are present. More recent research by Elias (2018) also finds no difference in gender attitudes to speeding fines. Notice that in both cases there are no strategic interactions, in the sense that the authors do not consider environments where less speeding by one group makes it easier for others to speed.”

19. Para 2, section 2, is your paper about cooperation?

It is more about behavior in social dilemmas, which we now make clearer, hopefully.

20. Page 6, Section2, para 2, line 4. Extra space before parenthesis. This repeats in several places.

We have carried out a global “search and replace” to deal with this issue.

21. Page 5, beginning, you state that, “Because this literature provides ambiguous messages about the effect of gender, we create a model in Section 3 that does not directly account for gender differences. Instead we allow the data to clarify the direction of effects (Section 5).” Then what is the usefulness of your model?

We thank the Reviewer for asking this question because it helps us make our point better. Our model is useful in the sense that we want to detect “anomalous” behavior according to some objective benchmark. As it turns out, just males behave in “unexpected” ways, and now we can point to this.

22. Page 6, para 2: “At this point” is unnecessary. It does not follow from a past argument.

Corrected.

23. Page 7, para 1. Remove “can”.

Done.

24. 3.1. The game: a. Explain better how the theoretical models maps into the experimental design. If you want the data to speak for itself, then how relevant is your theoretical framework?

As stated above in our reply to your point 21, we want to detect “anomalous” behavior according to some objective benchmark. As it turns out, just males behave in “unexpected” ways, and now we can point to this. Moreover, we have relegated the theory to an appendix, and just kept the hypotheses that discipline the analysis.

b. “driver” is out of place. Delete.

Done.

c. Define variables in pi.

Done.

d. The accident probabilities are pi. When defining EU on top of page 8, you state that ai defines accident probabilities?

We now make it clearer by saying that ai is a “idiosyncratic factor in accident probabilities”.

25. Section 3.2. a. Please clarify what you mean by imperfectly imposed fines. It becomes clearer later on that they are imposed on those that “are not in an accident.”

We now clarify that “This only affects participants who were not in an accident, since those in an accident already lose their whole experimental endowment.”

b. Endogenous. You have changed two things, the size of the fine which depends on how many vote for it. Better say that it is a variable sign that is increasing in the number of those that vote for the sign. Probably, accounting for beliefs probably controls for this in the analysis. From the results it seems that this should not matter. Discussing this would be useful.

We thank the Reviewer for this pertinent comment. We now clarify that “The probability `of the punishment is the same as in Exogenous but its size but its size depends on how many group members decide to punish the F choosing players.”

c. Endogenous: “But we social sanctioning…” incomplete sentence.

We now write “Social sanctioning has been shown to support mutual cooperation in large groups.”

d. From Proposition 2, can you rank the proportion of players choosing F for each treatment?

While we agree with the Reviewer that this is an interesting suggestion, we do not think we can do that unless we make more assumptions.

e. You define a variable alpha that captures sensitivity to fines. This could capture gender effects too.

The alpha variable is hardwired into the monetary experimental payoffs. There could be additional effects that are akin to those of the monetary alpha, but we do not think we can estimate them structurally with our data.

26. The results are very interesting. Endogenous and framing nudges work stronger for females. It would be interesting to see a discussion on why this is so. For example, brain research has shown that females are more consensus oriented and “aware” of their immediate surroundings (for example read The Female Brain, Louann Brizendine). These results also add to the literature on gender differences in cooperation and competition where they argue that context matters more for females (see, Gender Differences in Preferences, Rachel Croson and Uri Gneezy, Journal of Economic Literature, Vol. 47, No. 2, June 2009, pp. 448-74.

We thank the Reviewer again for considering that our results are interesting. We now have a full paragraph discussing the differential adherence to social norms of the genders in different scenarios, and we also discuss the possible origins, biological or social of those differences. One important difference is, as we note, “in our case males not just comply less, they take advantage of compliers to behave worse.”

 

Answers to Reviewer 2 and summary of changes made in response to his/her comments

Overall assessment

I have refereed this paper for a different journal and I must say that it was shocking to see that the authors had not taken on board the many comments they received from the editor and referees in that submission. This is a paper which tackles an interesting problem. The experimental design is clean. However, the theory section presents a number of problems and it is to a large extent irrelevant; critically, the theory has nothing to do with gender, despite gender is the main topic of the paper.

In addition to the problem with the theoretical model, the paper lacks focus, and feels like a collection of results. The authors remain surprisingly silent about why the policy interventions have the expected effect with females but encounter a backlash with males. This is a puzzle the paper does not tackle. The external validity of the exercise is already limited given the very specific behavior studied (driving) and the subject pool leveraged (university students, some of them surely not drivers). It makes the paper no favor that the authors do not attempt to rationalize this finding.

In addition, the paper is very meandrous. I am sure that there exists a shorter and more compact version of the paper conveying more effectively and clearly the main results of the experiment. But the paper, in its current form, is still a long way from there.

We thank the Reviewer for finding the problem we deal with interesting. We appreciate his/her comments and as we detail below we have tried our best to take them into account. We are sorry if the Reviewer believes we did not do that after receiving his/her feedback in previous occasions. We did try to accommodate all the input we receive although, unfortunately, it is often difficult to conciliate contradictory advice from different Reviewers and Editors. On the other hand, some of the suggestions may be matters of taste and/or factually inaccurate, and those we may not have followed. 

Theoretical model

In order to follow as closely as possible the Reviewer’s suggestions, we have indeed taken away the model from the core of the paper, leaving just the hypotheses, which as we explain to the other referee are useful because we want to detect “anomalous” behavior according to some objective benchmark. Having said that, we find it important nevertheless to respond to some of the Reviewer’s specific comments.

One caveat of the model is that the authors assume that changing from one speed to another has no effect on the average speed. This can be seen in pp. 7-8 where the average speed of players other than i remains defined as the average speed of the entire population. 

We respectfully disagree with the Reviewer here as we believe this assertion is not correct. The definition of AS_{-i} clearly states “Speed of the players other than i in the game is given by”, which is not the speed of the entire population and it is of course not affected by a change in the speed of i. The speed of the total population is then computed as (N-1)/N AS_{-i} + 1/N S¬_i, which is what we use throughout the proofs. 

The equilibrium concept is undefined and there is no explanation given in that regard. The proofs in the appendix show that the concept is likely to be the Nash equilibrium where agents choose the speed that maximize their utility given their beliefs about average speed that must be correct in equilibrium. There is no reason to avoid defining the equilibrium concept as the correctness of the theoretical results hinges critically on that; they cannot be correct or incorrect in a "theoretical vacuum". 

In this case we would like to apologize to the Reviewer because he/she is correct about this as we had not mentioned this point. In the new version we now clarify: “The risk preferences are the private information of each individual, and the equilibrium concept is Bayes-Nash.” In fact, this is a very relevant clarification because the private information is very important in relation to other comment from the Reviewer as we discuss below.

There are other issues related to the presentation of the model such as that the parameter ai appearing before being properly introduced; that the utility functions in p. 8 could be presented in a compact way using Di; 

We thank the Reviewer for noting these problems. These two issues have now been taken care of.

That there is no proof of Proposition 3 or at least a formal proof that a reduction in the payoff of F and AS is monotonic in fines and punishments; or that the proof or Proposition 1 assumes N = 10 without that being mentioned in the main text (the reader discovers that only when checking the proofs).

We once again apologize to the Reviewer, because we did forget to add N=10 in the description of the parameters and because we also forgot to include the proof of Proposition 3. 

Whilst these issues are just a nuisance, it is much more problematic that the main proposition, Proposition 1, and at least one of the hypotheses derived from it, Hypothesis 2, are wrong. These is a serious claim, so let me be very precise now.

Let me start with part 2 in Proposition 1. There it is said that there is no equilibrium where drivers only choose A or only F if the distribution of risk attitudes G(:) has positive mass everywhere in its support (0; 1): The proof in pp. 39-40 claims to prove that assertion by contradiction by showing

that for an agent to choose F when everybody else chooses F it must be that i 0:86848: Well, that is no contradiction. An equilibrium where every agent chooses F can happen with positive probability, actually with probability (1 G(0:86848))N if we assume risk parameters are i.i.d. draws from G(:): If all the draws are above that threshold, the all-Fast equilibrium exists. I have gone over this many times and I cannot see why the authors can state that the fact that i 0:86848 invalidates this. Choosing F when everybody else chooses F does not need to be a best response for all i for that to be an equilibrium. It is enough that it is a best response for all agents and there is a positive probability of that happening (the probability of the i of all players being above that threshold). It can be argued that such equilibrium is unlikely to happen (although that depends on the specific shape of G(:)) but what is clear is that Hypothesis 2, which is derived from this part of the proposition is wrong too; the theory does not lend support to the prediction that "participants in the experiment will never completely coordinate on choosing F". They may, with some positive probability.

In this point, we respectfully disagree with the Reviewer’s comment, although we acknowledge that his/her problem here might have arisen from lack of clarity in our writing. As mentioned above, the “The risk preferences are the private information of each individual, and the equilibrium concept is Bayes-Nash.” This means that if a (risk-preference) type of a player wants to deviate from the proposed equilibrium, this is indeed not a Bayes-Nash equilibrium. And that is precisely what we show, there are some types who want to deviate from the equilibria where every type of every player uses A or F. 

Unfortunately, part 1 of Proposition 1 is incorrect as well. The proof is not particularly helpful so consider instead the graph below. The black line corresponds to the locus in the parameter space ( i 2 (0; 1) is the risk attitude parameter, AS 2 [0:5; 2] is the belief about the average speed) where the utility of an agent driving slow is equal to the utility of that agent if they drove fast, that is EU(F) = 2 (1 0:35( 910AS + 210)) = 1 0:3( 910AS + 110) = EU(S) Parameters are such that EU(F) < EU(S) below that curve. Similarly, in the area below the red curve it holds that EU(A) = 0:5 < 1 0:3( 9

10AS + 110) = EU(S):

This demonstrates that the statement 1 in Proposition 1 ("There are no beliefs about AS and no value of i for which it is optimal to choose S") is false, as it is a best response to choose Slow for an agent whose i and beliefs of AS are in the area below the black curve. Consider for instance the case of an agent who expects everybody else to play F; i.e. AS = 2; if the agent is sufficiently risk averse, i.e., i < 0:5, their best response is indeed to play S: The intuition is clear and consistent with the strategic substitutes nature of the game.

Yet once again we respectfully disagree with the Reviewer, although we have to thank him/her as he/she provides a very nice alternative proof for the result in his/her comment. You claim that point below the red curve have the payoff of S bigger than the payoff of A. But it is easy to see that is incorrect. Take \\gamma =1 and AS=0.5. Then it is easy to see that the payoff of A is 1 and that of S is smaller than one. So, in the area above the red curve S > F and in the area below the black curve S > A, but since the two areas have no intersection, our result is in fact true.

All this forces me to conclude that the model should be scrapped from the paper. It adds very little, it is technically unsound and the way it models social norms is unhelpful in understanding the experimental findings.

As can be seen from our previous comments on the theoretical model, we do believe that our theory can be rigorously proven, but of course we agree with the Reviewer that the presentation should have been much better and we thank him/her for helping us making it clearer. In any event, we agree that a shorter paper can be useful and more widely read so we have moved the model to an appendix.

Presentation of the results

The presentation of the experimental results is plodding. I already mentioned in my previous report that the analysis of earnings was given too much prominence and could be scrapped. The authors kept it. And whilst I can see that as a matter of taste, I still believe that the current version hosts a proliferation of tables and results (11 tables in the main text, 7 questions and 7 results to a total of 34 pages excluding references). This detracts the paper from making its message come across effectively.

We have accepted the Reviewer’s suggestion and we have moved the analysis of earnings to the Supporting Information file. 

As mentioned above, the writing highlights how the behavior and beliefs of females is affected by the policies implemented. That is not the surprising result. After all, females behave as expected, they abide. They are affected by the policies and change their beliefs. It is the behavior of males what is

surprising. They not only do not reduce their speed but increase it. They do not adjust their beliefs. In sum, the results suggest strongly that the interventions emboldened males. This goes against all what should be expected and should, in my view, be highlighted much more. There is some material added about this compared to the last version I saw (in Section 5 and the conclusion), but it is featured only marginally (the main body of the paper remains intact) when it should be central.

We agree with the Reviewer that this is an important finding. In fact, it is a reason why having Hypothesis 3, derived from the theory is useful. We now highlight this result in the introduction, in the results section (both when discussing strategy choice and when discussing social norms) and in the conclusion.

Again, I believe that there is a better, shorter (and model free) version of the paper which is shorter and highlights its two main results: 1) Interventions do not reduce AS; 2) the reason is that whilst females respond as expected to the new policies, males are emboldened.

Let me also mention that I am not entirely convinced that the data supports Result 5. The interaction FramingxFemale in Table 10 is not significant. The coefficients suggest rather that Framing has an effect but that is not gender-specific.

In order to take this suggestion from the Reviewer into account, we now moderate the statement of Result 5 “In the presence of the $Endogenous$ policy condition, female participants are more likely to state Auto as their personal normative belief instead of Fast.”

Minor comments

1. There are several citations missing (e.g. p.1) and figures not correctly

referenced (e.g. p. 15).

As we answered to Reviewer 1, we do not know what is the reason, but we have not been able to nail down this problem because we do not observe any of this in our manuscript. Perhaps it was a product of the compilation of the file in the journal server? In any event, for this new version we are sending a pdf file making sure that there are no missing references.

2. There is no mention to when the experiment was run.

We are sorry we did not include this information in the previous version of the paper. We now explain that “Experiments were conducted between January and May of 2018 at University College

London’s Experimental Laboratory for Finance and Economics.”

3. That the Holt and Laury method was used to elicit risk preferences would probably be better placed in p. 11-12 when describing the experimental procedures.

Done.

4. Table 14 shows that participants in the Exogenous treatment had different risk preferences than the rest. Could that explain that gender differences in speed are only weakly significant under that policy? (in contrast with the other policies, where gender differences are much stronger).

We thank the Reviewer for the suggestion. We now say “Note, though, that the subject pool is a bit more risk-averse on average (although the difference only has a p-value of 0.06). This could perhaps partially explain the dampened results in Exogenous.”

5. It would be nice to see a comment on why Endogenous is the treatment that generates the lightest sway in beliefs.

We are afraid that we are not entirely sure what the Reviewer means here, and therefore we have taken no action. Endogenous is in fact the treatment which generates the largest change in beliefs, especially for females. Perhaps we misunderstand the question?

6. It would be useful for the reader to have a footnote explaining how guesses of the driving distributions translated into earnings (pp. 11-12).

In regard to this question, we would like to point out that this was already stated in the paper in p. 12 of the original manuscript, and we have kept the sentence in the new one. The sentence is: “The amount a participant earned from their guess was 5 minus the difference between their guessed distribution of driving styles and the actual distribution of driving styles in that round. A perfect guess earned 5 and a very inaccurate guess earned 0.”

7. Why the regressions on driving choices (Table 4) do not include the

previous’ round decision as a control? 

About this suggestion, we have to say that in an earlier version of the manuscript we had included this variable in the regressions, but in the reviewing process another Reviewer asked us to remove that variable, and we have left it like that. 

8. Why is the proposed designed a "mixed-agency" scenario? (p.. 33). This concept is unexplained.

We have removed this unexplained concept and its corresponding sentence from the manuscript. 

9. The authors have not bothered correcting the typos already pointed out in submission this referee already evaluated: p. 4, duplicated "However"; p. 4 "being" instead of "been"; p. 10, "Endogeno" instead of "Endogenous"; p. 20, a "to" is missing after "likely"; p. 25, "participants"; p. 30, a "to" is missing after "likely".

We apologize to the Reviewer for this oversight. These are now corrected, with the exception of p. 25, "participants", which we have not been able to find.

---

## [Decision Letter · Decision Letter 1]

20 Jul 2022

PONE-D-22-09104R1The effectiveness of prosocial policies: Gender differences arising from social normsPLOS ONE

Dear Dr. Sánchez,

Thank you for submitting your manuscript to PLOS ONE. After careful consideration, I feel that it has significantly improved but does not fully adress major criticism received by Reviewer 2. Therefore, I invite you to submit a revised version of the manuscript that addresses the points raised by Reviewer 2.

We look forward to receiving your revised manuscript.

Kind regards,

Nikolaos Georgantzis, Dr.

Academic Editor

PLOS ONE

Reviewers' comments:

Reviewer's Responses to Questions

**Comments to the Author**

1. If the authors have adequately addressed your comments raised in a previous round of review and you feel that this manuscript is now acceptable for publication, you may indicate that here to bypass the “Comments to the Author” section, enter your conflict of interest statement in the “Confidential to Editor” section, and submit your "Accept" recommendation.

Reviewer #1: All comments have been addressed

Reviewer #2: (No Response)

2. Is the manuscript technically sound, and do the data support the conclusions?

Reviewer #1: Yes

Reviewer #2: Partly

3. Has the statistical analysis been performed appropriately and rigorously? 

Reviewer #1: Yes

Reviewer #2: Yes

4. Have the authors made all data underlying the findings in their manuscript fully available?

Reviewer #1: Yes

Reviewer #2: Yes

5. Is the manuscript presented in an intelligible fashion and written in standard English?

Reviewer #1: Yes

Reviewer #2: Yes

6. Review Comments to the Author

Reviewer #1: The authors have addressed all comments. The paper is clearer and reads much better now. Some tables and figures seem to be misaligned. I assume this can be addressed in the final stage.

Reviewer #2: I have read with interest this revised version of the manuscript. My overall assessment is that the revision has gone in the right direction but there is still work to be done. The reason is partially that the authors have not incorporated significant comments made in previous reports.

To start with the positives: the paper now makes a commendable effort in addressing the problem of external validity and in portraying correctly the results. The authors now include interesting considerations showing that the behavior studied in this paper, despite being framed in a driving context, is relevant to other behaviors with externalities. It seems that, finally, the authors have taken seriously the suggestion that it is not that the behavior of women is anomalous (actually it is very consistent and rational) but the behavior of men. It is also commendable that the authors have shortened the empirical analysis which is now more direct and to the point. Most formal issues with the paper have now been corrected.

On the negative side, I am again shocked that the authors insist in keeping the model. In their response to this referee they mentioned that "we have moved the model to an appendix". However, a description of the model can still be found in p. 4-5. This is not a matter of taste. Leaving aside its correctness and clarity, the model does not help the reader in any way. It actually detracts from the paper. The model has only one source of individual heterogeneity: risk aversion. This alone cannot explain the gender differences observed in the paper. The model cannot explain either the most interesting result in the paper: that men free-ride on women responding to the policies/incentives provided. In other words, the model sheds no light on why "the outcomes may vary for different groups of the population" (p.4). The predictions of the model are either common-sensical (hypothesis 3) or could be obtained with other models and do not actually hold in the lab. The Endogenous treatment could be implemented in the model but it is not... To be honest, it is beyond me why the authors still want to keep such model despite the arguments provided by several referees, including this one. The current version of the paper is some kind of compromise but in addition to the above it is unsatisfactory because it actually leads to a duplicated presentation of treatments (in p. 5 and p. 7). The experiment and its treatments can be presented without the model. So there is really no excuse for keeping it.

Other comments:

There are some other loose ends. In addition, the paper could still improve on the formal side.

- The results driving in the previous literature summarized in p. 3 lines 62-66, seem to run against the ones in the paper. This in itself is no problem. However, the authors describe these results as not relevant to theirs since they are obtained in the absence of strategic interactions. This is not entirely correct. Strategic interactions are always present in driving. Maybe in those works cited strategic interactions are less intense because the population of drivers is larger than in the experiment, but certainly they remain.

- Another proof that the current compromise in the model is not satisfactory is the derivatives in p.4. What are they for? What do they imply? By the way, the last gamma in the first derivative seems to be missing a subscript.

- The description of the experiment has improved but there is still room for further improvement. There is information still missing on session duration, average earnings, structure of payments (e.g. show up fee), etc.

- The endogenous treatment seems the most effective and the one where the strongest differences emerge. It seems surprising that the authors do not exploit it more. For instance, there is no discussion on the contribution decisions. Maybe the belief about how many people contributed to the punishment fund were different between genders. The same with the actual choices. If, as the authors suggest in lines 487-8, "female participants would be more responsive to realizing a violation of their (non-Fast) norm", should not they also contribute more to the punishment fund? Sadly, these possibilities remain unexplored.

- I am not sure the study of decisions after being fined is very convicing. The reason is that females who drive Fast might be different from Males who do the same. Fines are not allocated randomly. Because of that, it is not that surprising that males and females respond differently. If this analysis is to be kept this possibility must be mentioned or proof should be provided that the two sub-samples are comparable in observables.

- Does the result about males in Result 4 suggest that males anticipate that there will be a backlash against the policy?

- It seems a bit weird to state that "our design also allows us to identify the mechanism driving these differences in behavior" (lines 555-7) after having stated "we cannot make a clear identification of the mediator at work here" (lines 533-534). It seems that the toning down of certain statements has not been applied consistently throughout the paper.

Minor comments

- "ex ante primes" (p.2). I'm not sure this is correct English. Priming seems more correct than "primes". In any case, all priming is ex ante (otherwise it would be called nudging or some other thing), so the expression seems redundant.

- Similarly, the expression "the average effect on genders" (e.g. lines 521-2) sounds a bit weird. Something like "the differential effect of policies by gender" seems more correct.

- The "or" in Hypothesis 3 can be safely replaced by an "and"

- There is often a mix up of present and past tense to describe the experiment (for instance, lines 225-235). The convention is to use the past tense but what consistency is more important; either one or the other.

- The phrasing "participants who chose Fast had a 25% chance to pay a fine" is not very precise and a bit weird. "Participants who chose Fast were fined with £4 with a 25% chance each round" seems more precise.

- Result 3 is mentioned in line 358 as if it had been already stated but it has not.

- Probably as a remainder of previous versions, lines 358-360 characterize the choices of females as puzzling when it is actually males' decisions the ones really puzzling.

- The use of beliefs to explain the results on AS would probably be better placed earlier to explain the importance of the exercise. At the moment, the first paragraphs in the "Empirical expectations" section leave the reader without an explanation of why the exercise is worth doing.

- I found very little differences between Question 1 & 2 on the one hand and Question 3 & 4 on the other. Maybe they could be integrated in just one question each.

- The verb in the sentence in line 540 is inconsistent: "The origin of that gender differential compliance ARE still underexplained"

7. PLOS authors have the option to publish the peer review history of their article (what does this mean?). If published, this will include your full peer review and any attached files.

Reviewer #1: No

Reviewer #2: No

---

## [Author Response · Author response to Decision Letter 1]

26 Aug 2022

Reviewer #2: 

I have read with interest this revised version of the manuscript. My overall assessment is that the revision has gone in the right direction but there is still work to be done. The reason is partially that the authors have not incorporated significant comments made in previous reports.

We thank the Reviewer for his/her appreciation of our new manuscript. 

On the negative side, I am again shocked that the authors insist in keeping the model. In their response to this referee they mentioned that "we have moved the model to an appendix". However, a description of the model can still be found in p. 4-5. This is not a matter of taste. Leaving aside its correctness and clarity, the model does not help the reader in any way. It actually detracts from the paper. The model has only one source of individual heterogeneity: risk aversion. This alone cannot explain the gender differences observed in the paper. The model cannot explain either the most interesting result in the paper: that men free-ride on women responding to the policies/incentives provided. In other words, the model sheds no light on why "the outcomes may vary for different groups of the population" (p.4). The predictions of the model are either common-sensical (hypothesis 3) or could be obtained with other models and do not actually hold in the lab. The Endogenous treatment could be implemented in the model but it is not... To be honest, it is beyond me why the authors still want to keep such model despite the arguments provided by several referees, including this one. The current version of the paper is some kind of compromise but in addition to the above it is unsatisfactory because it actually leads to a duplicated presentation of treatments (in p. 5 and p. 7). The experiment and its treatments can be presented without the model. So there is really no excuse for keeping it.

We have now removed completely the model. 

Other comments:

There are some other loose ends. In addition, the paper could still improve on the formal side.

- The results driving in the previous literature summarized in p. 3 lines 62-66, seem to run against the ones in the paper. This in itself is no problem. However, the authors describe these results as not relevant to theirs since they are obtained in the absence of strategic interactions. This is not entirely correct. Strategic interactions are always present in driving. Maybe in those works cited strategic interactions are less intense because the population of drivers is larger than in the experiment, but certainly they remain.

We thank the reviewer for making us note this point. In order to be more precise, we now write: “A difference with our framework is that in the latter two cases the authors consider environments where less speeding by one group does not makes it easier for others to speed.”

- Another proof that the current compromise in the model is not satisfactory is the derivatives in p.4. What are they for? What do they imply? By the way, the last gamma in the first derivative seems to be missing a subscript.

As the model has been removed, this part has been removed as well. 

- The description of the experiment has improved but there is still room for further improvement. There is information still missing on session duration, average earnings, structure of payments (e.g. show up fee), etc.

We now write in the experimental design section: “Each experiment session lasted approximately 90 minutes and none of them lasted no longer than 2 hours. The average payment was 23.14 GBP. The maximum payment was 43.9 and the minimum was 5.1. These numbers include a 5GBP showup fee.”

- The endogenous treatment seems the most effective and the one where the strongest differences emerge. It seems surprising that the authors do not exploit it more. For instance, there is no discussion on the contribution decisions. Maybe the belief about how many people contributed to the punishment fund were different between genders. The same with the actual choices. If, as the authors suggest in lines 487-8, "female participants would be more responsive to realizing a violation of their (non-Fast) norm", should not they also contribute more to the punishment fund? Sadly, these possibilities remain unexplored.

Thanks for the suggestion. First, regarding the question whether the belief about other participants’ contribution to the punishment fund differed by gender. Unfortunately, we cannot answer it because we do not have those data. We did not ask for the participants’ beliefs about others’ contributions to the punishment fund. On the other hand, we have looked at the contribution decisions by gender, and we now write just after Result 4 that “One interesting question in this context is whether the gender differences in normative beliefs about driving choices extends to the imposition of fines in the Endogenous policy condition. We did not ask a question about norms in the case of fines, but the actions do not differ very strongly. Female participants contribute to the punishment fund 13.5% of the time, which is higher than male participants (8.6%). However, these differences are based on only 8 sessions of the Endogenous treatment and a logit model which clusters at the session level shows this difference to be insignificant (details in the SI file).” The analysis of the punishment decisions was already in the SI file in the previous version, but we thought it was too detailed for the main text.

- I am not sure the study of decisions after being fined is very convincing. The reason is that females who drive Fast might be different from Males who do the same. Fines are not allocated randomly. Because of that, it is not that surprising that males and females respond differently. If this analysis is to be kept this possibility must be mentioned or proof should be provided that the two sub-samples are comparable in observables.

We have eliminated this analysis.

- Does the result about males in Result 4 suggest that males anticipate that there will be a backlash against the policy?

While we believe that the Reviewer’s question is a very interesting one, Result 4 simply says females think the norm will change to Auto under Endogenous and males think it will shift to Auto in Framing. We are not entirely sure that this suggests males anticipate a backlash and therefore we have chosen to leave this point out of the discussion.

 - It seems a bit weird to state that "our design also allows us to identify the mechanism driving these differences in behavior" (lines 555-7) after having stated "we cannot make a clear identification of the mediator at work here" (lines 533-534). It seems that the toning down of certain statements has not been applied consistently throughout the paper.

We now say “Our design allows us to suggest possible mechanisms driving these differences in behavior, by using question about personal normative beliefs and empirical and normative expectations to elicit social norm”.

Minor comments

- "ex ante primes" (p.2). I'm not sure this is correct English. Priming seems more correct than "primes". In any case, all priming is ex ante (otherwise it would be called nudging or some other thing), so the expression seems redundant.

We now say “which raises the question as to whether priming may be useful.”

- Similarly, the expression "the average effect on genders" (e.g. lines 521-2) sounds a bit weird. Something like "the differential effect of policies by gender" seems more correct.

We have adopted the Reviewer’s suggestion. 

- The "or" in Hypothesis 3 can be safely replaced by an "and"

Done.

- There is often a mix up of present and past tense to describe the experiment (for instance, lines 225-235). The convention is to use the past tense but what consistency is more important; either one or the other.

Done.

- The phrasing "participants who chose Fast had a 25% chance to pay a fine" is not very precise and a bit weird. "Participants who chose Fast were fined with £4 with a 25% chance each round" seems more precise.

Done.

- Result 3 is mentioned in line 358 as if it had been already stated but it has not.

Result 3 was not needed there, and on the other hand we have merged it with Result 4 as per the Reviewer’s suggestions, so we have deleted this mention.

- Probably as a remainder of previous versions, lines 358-360 characterize the choices of females as puzzling when it is actually males' decisions the ones really puzzling.

We have deleted the mention of their choices as puzzling.

- The use of beliefs to explain the results on AS would probably be better placed earlier to explain the importance of the exercise. At the moment, the first paragraphs in the "Empirical expectations" section leave the reader without an explanation of why the exercise is worth doing.

We have moved the sentence “Following Bicchieri (2006), a social norm exists in a group if a majority of members share empirical expectations (beliefs about what most others in the group will do) and normative expectations (beliefs about what most others in the group believe one should do)” to the beginning of the section on empirical expectations.

- I found very little differences between Question 1 & 2 on the one hand and Question 3 & 4 on the other. Maybe they could be integrated in just one question each.

We have done as suggested. 

- The verb in the sentence in line 540 is inconsistent: "The origin of that gender differential compliance ARE still underexplained"

This has been corrected.

---

## [Decision Letter · Decision Letter 2]

13 Sep 2022

PONE-D-22-09104R2The effectiveness of prosocial policies: Gender differences arising from social normsPLOS ONE

Dear Dr. Sánchez,

Thank you for submitting your manuscript to PLOS ONE. As the journal allows for minor changes but not conditional acceptance, I am obliged to ask for minor changes. Please make the changes suggested by the reviewer below and resubmit, in order to finally accept your paper. Please submit your revised manuscript by Oct 28 2022 11:59PM. If you will need more time than this to complete your revisions, please reply to this message or contact the journal office at plosone@plos.org. If applicable, we recommend that you deposit your laboratory protocols in protocols.io to enhance the reproducibility of your results. Protocols.io assigns your protocol its own identifier (DOI) so that it can be cited independently in the future. For instructions see: https://journals.plos.org/plosone/s/submission-guidelines#loc-laboratory-protocols. Additionally, PLOS ONE offers an option for publishing peer-reviewed Lab Protocol articles, which describe protocols hosted on protocols.io. Read more information on sharing protocols at https://plos.org/protocols?utm_medium=editorial-email&utm_source=authorletters&utm_campaign=protocols.

We look forward to receiving your revised manuscript.

Kind regards,

Nikolaos Georgantzis

Academic Editor

PLOS ONE

Journal Requirements:

Reviewers' comments:

Reviewer's Responses to Questions

**Comments to the Author**

1. If the authors have adequately addressed your comments raised in a previous round of review and you feel that this manuscript is now acceptable for publication, you may indicate that here to bypass the “Comments to the Author” section, enter your conflict of interest statement in the “Confidential to Editor” section, and submit your "Accept" recommendation.

Reviewer #2: All comments have been addressed

2. Is the manuscript technically sound, and do the data support the conclusions?

Reviewer #2: Yes

3. Has the statistical analysis been performed appropriately and rigorously? 

Reviewer #2: Yes

4. Have the authors made all data underlying the findings in their manuscript fully available?

Reviewer #2: Yes

5. Is the manuscript presented in an intelligible fashion and written in standard English?

Reviewer #2: Yes

6. Review Comments to the Author

Reviewer #2: I was happy to see that the authors addressed successfully all the issues I mentioned in my previous report.

Only a few minor formal comments remain.

The most important one is about the abstract: As the authors say in their response to my previous report, they have removed the model completely from the paper. However, the abstract does not reflect that. It mentions the model quite a few times. The authors should revise the abstract so it reflects the new reality of the paper and its contents.

- Line 42: It should probably read "policies" rather than "policy"

- Lines 151-152: This sentence has a double negation that makes it mean the opposite of what the authors intend. Delete the "no" in line 152.

- Lines 161-162: In purity, it is not true that subjects could not know which one was the last round. Any subjects who reached round 25 did. The authors could mention this in #footnote 8 and the instances when this happened (probably zero)

- Page 6: The authors may want to mention that having just three actions has the advantage of facilitating belief elicitation. Just a suggestion.

-Lines 220 and 224. To be consistent with the notation introduced earlier in the paper, it may make sense to include the notation C and write C=£4 and C=£X.

- Lines 305-306: Change either "magnitude" to "magnitudes" or "are" to "is".

- Line 347: Seems there is no space before "As".

7. PLOS authors have the option to publish the peer review history of their article (what does this mean?). If published, this will include your full peer review and any attached files.

Reviewer #2: No

---

## [Author Response · Author response to Decision Letter 2]

14 Sep 2022

All minor changes indicated by the Reviewer have been implemented

---

## [Editor Report · Decision Letter 3]

15 Sep 2022

The effectiveness of prosocial policies: Gender differences arising from social norms

PONE-D-22-09104R3

Dear Dr. Sánchez,

We’re pleased to inform you that your manuscript has been judged scientifically suitable for publication and will be formally accepted for publication once it meets all outstanding technical requirements.

Kind regards,

Nikolaos Georgantzis

Academic Editor

PLOS ONE
---

## [Editor Report · Acceptance letter]

12 Oct 2022

PONE-D-22-09104R3 

The effectiveness of prosocial policies: Gender differences arising from social norms 

Dear Dr. Sánchez:

I'm pleased to inform you that your manuscript has been deemed suitable for publication in PLOS ONE. Congratulations! Your manuscript is now with our production department. 

Kind regards, 

on behalf of

Prof. Nikolaos Georgantzis 

Academic Editor

PLOS ONE